# Advancing aerogel recyclability through polyhexahydrotriazine reactivity

Chang-Lin Wang [1], Yi-Ru Chen [1], Brahim Mezari[2], Fabian Eisenreich [1] & Željko Tomović [1] ✉

Organic aerogels have emerged as promising materials for advanced thermal insulation. However, their chemically robust covalent networks pose a major barrier for effective recycling. Introducing specific chemical bonds into the aerogel scaffold that enable on-demand reversibility offers a viable pathway to enhance recyclability and promote the sustainable use of these materials. In this work, we demonstrate that hexahydrotriazine (HT) units undergo nucleophilic attack by amines and engage in metathesis reactions, fundamentally redefining their reactive behaviors. Based on this, we introduce a waste-minimized, closed-loop chemical recycling process for highly porous, thermally superinsulating organic aerogels. These materials are partially depolymerized into soluble oligomers upon exposure to primary amines and can be reassembled into fresh polymer networks on demand. Additionally, by varying the amine feedstocks during depolymerization, we tailor key aerogel properties, such as thermal conductivity and flame resistance, beyond their initial synthesis. Under heat and pressure, HT bond exchange enables aerogels to transform into high-performance thermoset-like films, which subsequently revert to aerogels. This breakthrough in HT chemistry sets a benchmark for atom-efficient recycling, reprogramming, and reprocessing of HT-based materials, providing a transformative foundation for a circular materials platform with broad impact.

Aerogels, identified by IUPAC as one of the top ten emerging technologies in chemistry, present a promising material for energy conservation[1]. These materials exhibit ultralow thermal conductivity, reaching values lower than $0.020\,\mathrm{Wm^{-1}K^{-1}}$, far surpassing the insulating performance of conventional materials, such as polystyrene foams, polyurethane foams, and glass wool[2,3]. Organic aerogels, in particular, have shown great promise for reducing energy consumption and greenhouse gas emissions due to their ultralow density, great mechanical properties, and exceptional thermal performance[3,4]. While addressing these key challenges in sustainable development is essential, ensuring the recyclability of organic aerogels is equally important to establish them as truly sustainable materials. However, the highly crosslinked and robust molecular structures of traditional organic aerogels render them non-recyclable. To address this limitation, it is imperative to integrate recyclability as a core criterion in the design of new organic aerogel materials, all while maintaining their exceptional high-performance properties, an ambitious yet critical scientific challenge. One promising approach involves using crosslinked networks with reversible chemical linkages, such as imines and hexahydrotriazines (HT), to achieve closed-loop recycling under mild conditions[5–7]. In these systems, straightforward hydrolysis reactions enable selective and efficient depolymerization, reverting the materials to their original monomeric building blocks. These monomers can be isolated with high purity and reused directly to synthesize the same

[1]Polymer Performance Materials Group, Department of Chemical Engineering and Chemistry and Institute for Complex Molecular Systems (ICMS), Eindhoven University of Technology, Eindhoven, The Netherlands. [2]Inorganic Materials and Catalysis Group, Department of Chemical Engineering and Chemistry, Eindhoven University of Technology, Eindhoven, The Netherlands. ✉e-mail: z.tomovic@tue.nl

organic aerogel materials, ensuring a circular recycling process. Yet, isolating monomers after depolymerization involves waste-generating purification steps. To eliminate these labor-intensive processes and prevent the loss of valuable chemical resources, such as monomers, acids, bases, and solvents, we recently introduced an unprecedented recycling strategy for organic aerogels, termed the aerogel-to-sol-to-aerogel (ASA) process[8]. Instead of capitalizing on reversible chemical bonds, this method relies on reshuffling and reorganizing covalent bonds integrated in organic aerogel networks. Using aminolysis with additional amines, we demonstrated that fully crosslinked polyimine aerogels can be efficiently broken down into soluble oligomers, which can then be repolymerized by introducing an appropriate amount of suitable aldehydes. This unique foam-to-foam recycling approach utilizes 100% of the originally building blocks and avoids external purification steps, representing a significant advancement in the

environmentally friendly and resource-efficient management of thermally insulating materials.

In our efforts to adapt the ASA process to different types of aerogel networks, we made an unexpected discovery: HT moieties, formed through the condensation of formaldehyde and primary amines[9], display reactivity through aminolysis with other primary amines, showcasing their ability to undergo nucleophilic attacks under mild conditions, and metathesis with other HT groups. Traditionally, HT groups are known to undergo nucleophilic attacks only with thiols and phosphines under harsh conditions, making them unsuitable for chemical recycling strategies through the ASA process[10,11]. Building on this groundbreaking discovery, we developed fully recyclable poly-hexahydrotriazine (PHT) aerogels (Fig. 1a). The aminolysis reaction between HT groups and aromatic primary amines enables the cleavage of crosslinked PHT networks into soluble aminal oligomer mixtures.

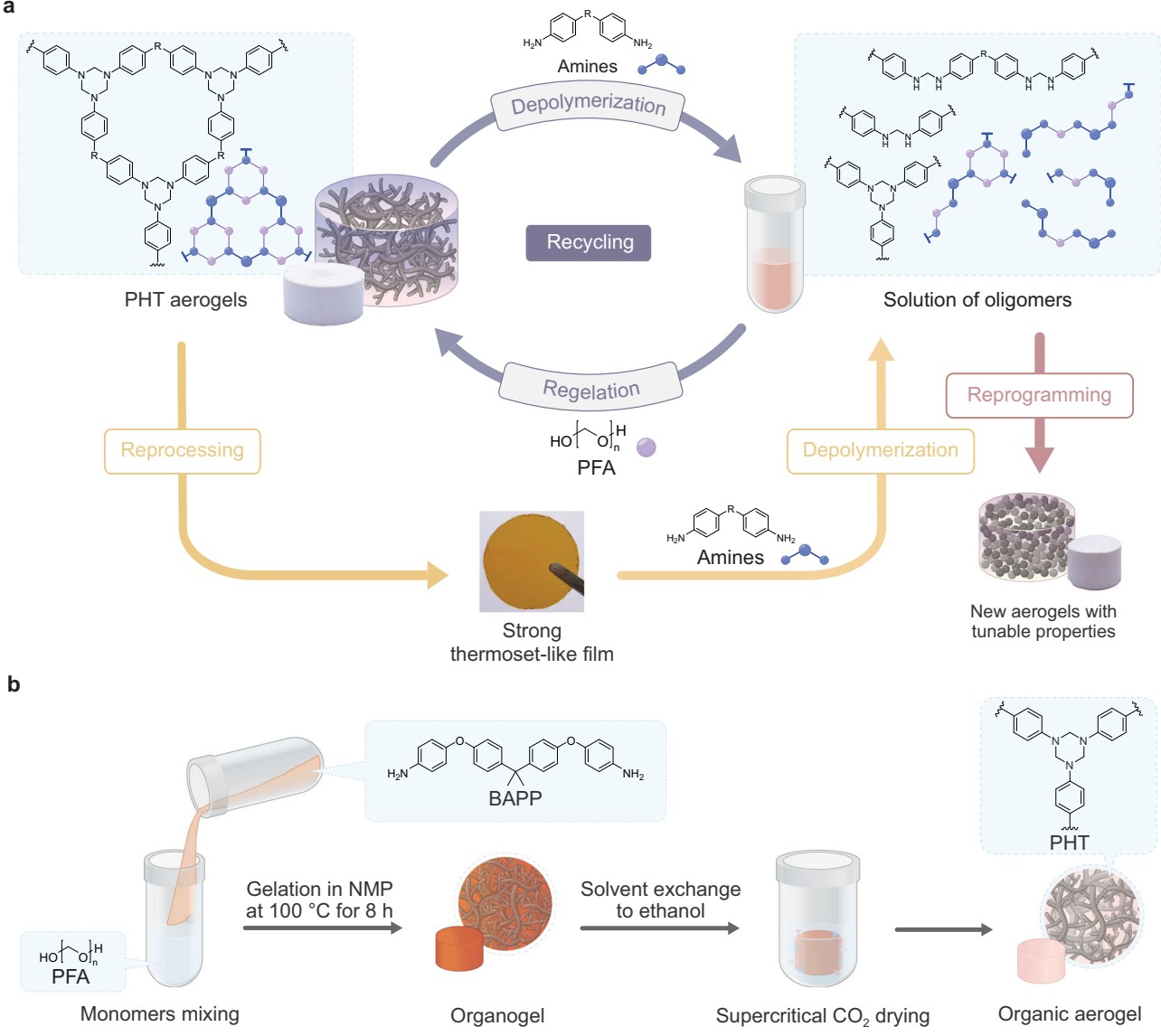

**Fig. 1 | The recycling process and synthesis of PHT aerogel. a** Schematic representation of the ASA process. Through the addition of primary amines, the cross-linked PHT network is partially depolymerized into soluble oligomer mixtures that can be reconnected by introducing an equivalent amount of repeating units in PFA, achieving aerogel recycling. By utilizing different types of amines, PHT aerogel can also be reprogrammed into new aerogels with tunable properties. In addition, PHT aerogels can be processed into strong thermoset-like films that can also be partially

depolymerized into a solution of oligomers with the introduction of amines. The regelation can be further initiated by introducing an equivalent amount of repeating units in PFA to prepare fresh aerogel once again. **b** Schematic representation of the aerogel synthesis protocol. Precursors were dissolved separately in NMP, mixed at 100 °C for 8 h, and cured overnight at room temperature to obtain stable organogels. The NMP solvent was removed by solvent exchange to ethanol before supercritical drying with $CO_2$ was applied to prepare PHT-pristine.

The soluble mixtures can be directly used to synthesize new aerogels by incorporating an equimolar amount equivalent amount of repeating units in paraformaldehyde (PFA). This process facilitates the continuous recycling of PHT aerogels without the need for additional purification, ensuring that all chemical resources are preserved. Remarkably, the newly synthesized aerogels maintain their nanoscale porous structure and thermal insulation properties for at least two recycling cycles. Furthermore, the reactivity of HT moieties offers significant tunability in the aerogels' properties. By incorporating specific aromatic amines, the material can be repurposed in every cycle on demand to enhance the thermal conductivity, mechanical strength, or thermal/flame resistance of the next-generation aerogel. In addition, metathesis exchange between different PHT structures facilitates the transformation of PHT aerogels into compact thermoset-like films through compression molding. These films demonstrate pronounced thermal stability and great mechanical robustness, which can be seamlessly reverted to their original aerogel forms using the ASA process (Fig. 1a). This circular lifecycle, driven by the unexplored reactivity of HT moieties, provides a sustainable pathway for developing recyclable high-performance polymer materials.

## Results and discussion
### Reactivity of HT structures

To validate the ability of HT structures to undergo aminolysis with primary amines, we first conducted small-molecule model reactions. Specifically, we selected 1,3,5-tris(4-methoxyphenyl)−1,3,5-triazinane (OMeHT) as a model HT compound, synthesized from p-anisidine and formaldehyde in the form of PFA, and subjected it to a reaction with an excess of p-anisidine (4.5 equivalents) to initiate an aminolysis reaction (Supplementary Fig. S1a). The progress of the reaction was monitored over time using $^1$H NMR spectroscopy in dimethylsulfoxide-$d_6$ (DMSO-$d_6$) at 65 °C (Supplementary Fig. S1b). Within 4 h, a decrease in the signal intensity of the HT compound and the emergence of characteristic peaks at 4.3 and 5.5 ppm, corresponding to the aminal structure formed via aminolysis, were observed (Supplementary Fig. S1b). According to the kinetic profile, the aminolysis reached an equilibrium at 59% consumption of OMeHT, resulting in 36% formation of the corresponding aminal (Supplementary Fig. S2). The reaction mixture was subsequently converted back into pure OMeHT by adding an equivalent amount of repeating units in PFA relative to the amine groups present in the reaction mixture (Supplementary Figs. S1c and S3). This experiment demonstrates the aminolysis and breakdown of HT moieties in the presence of additional aromatic amines. Importantly, this process is reversible, forming the foundation of the ASA recycling process when applied to polymeric materials. Encouraged by these findings, we proceeded to investigate whether bond exchange reactions could also occur between HT structures through a metathesis mechanism under heat and pressure. A model experiment was conducted by hot-pressing equimolar amounts of OMeHT and its ethoxy counterpart, 1,3,5-tris(4-ethoxyphenyl)−1,3,5-triazinane (OEtHT). The process involved applying 12 MPa of pressure at 180 °C for 0.5 h, followed by 40 MPa of pressure at 180 °C for an additional 0.5 h. The resulting solid mixture was analyzed by $^1$H NMR spectroscopy, which confirmed the structural integrity of the HT compounds without evidence of irreversible degradation, verifying their thermal stability (Supplementary Fig. S4a and S5). Since $^1$H NMR spectroscopy cannot distinguish between HT structures with exclusive methoxy or ethoxy substituents, and a combination of both, we turned to MALDI-TOF mass spectrometry for further analysis (Supplementary Fig. S4b). As anticipated, we not only identified the starting HT compounds but also detected HT molecules with varying ratios of methoxy and ethoxy substituents, confirming that reshuffling of the aromatic amines occurred under these conditions.

### Synthesis and closed-loop recycling of PHT aerogels

To apply the reactivity of HT moieties for organic aerogel recycling, we synthesized a PHT aerogel through the polycondensation of 2,2-bis[4-(4-aminophenoxy)phenyl]propane (BAPP) and PFA (Fig. 1b)[7]. To ensure the exclusive formation of HT structures and prevent the generation of unwanted hemiaminal intermediates—which could compromise the thermal stability and mechanical performance of the final materials—it is essential to react an equimolar amount of repeating units in PFA and amine[7,12,13]. The two precursors were mixed in N-methylpyrrolidone (NMP) at 100 °C for 8 h, resulting in the formation of a stable organogel, which was further cured at room temperature for 24 h (Fig. 1b). After being solvent exchanged to ethanol, the organogel was dried using supercritical $CO_2$, preserving the porous structure of the aerogel network. To enhance the sustainability of the PHT synthesis, all solvents used were recovered through stepwise distillation with high purity and good to high recovery yields (Supplementary Fig. S6 and Supplementary Table S1). The resulting PHT aerogel (PHT-pristine) exhibits low bulk density (0.15 gcm$^{-3}$) and high porosity (88%), key features of lightweight macro-mesoporous materials. PHT-pristine also shows a large specific surface area of 129 m$^2$g$^{-1}$, a total pore volume of 6.08 cm$^3$g$^{-1}$, and a mesopore volume of 0.25 cm$^3$g$^{-1}$, indicative for its highly porous architecture. It is noted that PHT-pristine possesses a predominantly macroporous architecture with a mesoporous fraction as measured by nitrogen sorption (Supplementary Fig. S7a). Importantly, PHT-pristine exhibits an ultralow total thermal conductivity of 18.8 mW m$^{-1}$K$^{-1}$ (Supplementary Table S2). Applying the Knudsen model shows that the gaseous contribution to the total conductivity is 9.3 mW m$^{-1}$K$^{-1}$, which is significantly lower than that of open still air, while the remaining 9.5 mW m$^{-1}$K$^{-1}$ originates from the solid phase[14]. These results indicate that the excellent thermal insulation performance of PHT-pristine arises from the interplay of Knudsen-induced reduction in gas-phase conduction and inherently low solid-phase conductivity. Given its exceptional material performance as a thermal insulating unit, PHT-pristine serves as a promising starting point for the development of a sustainable aerogel system. Building on the results of the model reactions, we explored whether PHT-pristine aerogel undergoes aminolysis in the presence of primary amines. This process would result in partial depolymerization into soluble oligomeric structures and a drastic reduction in crosslink density (Fig. 1a). To initiate the aerogel depolymerization process, PHT-pristine fragments were treated with a 14 wt% BAPP solution in NMP, using an amount equivalent to 1.5 times the BAPP content present in the aerogel. This amount was found to be the minimum quantity of BAPP necessary for dissolution of the PHT aerogel (Supplementary Fig. S8). After 4 h of ultrasonication, a clear red solution containing soluble oligomeric structures was obtained. The mixture was firstly analyzed by NMR spectroscopy. In the $^1$H NMR spectrum, characteristic signals for aminal structures were detected at 4.5 and 6.3 ppm (Supplementary Fig. S9a). The $^1$H COSY NMR spectrum further reveals that these two signals are coupled, providing further confirmation of the embedded aminal structure (Supplementary Fig. S9b). Moreover, the DEPT $^{13}$C NMR spectrum shows a methylene carbon signal at 54 ppm, corresponding to the aminal groups and aligning with similar aminal-containing structures reported in the literature (Supplementary Fig. S9c)[15]. Similarly, the aminal structure was also identified by MALDI-TOF analysis, where the aminal group is presented as a BAPP dimer (Supplementary Fig. S10). To confirm the presence of the formed oligomers, $^1$H DOSY NMR experiments were conducted. The spectra revealed that the oligomeric structures exhibiting the characteristic aminal signals are larger in size than the one of BAPP amine (Supplementary Fig. S11a). In addition, this oligomer mixture was analyzed using Gel Permeation Chromatography (GPC) in N,N-dimethylformamide (DMF), which showed a broad molecular weight distribution ranging from 400 to 7000 Da (Supplementary Fig. S11b).

Overall, these findings align with our model reaction, strongly indicating that nucleophilic attacks occur within the polymer system. Moreover, they confirm that the PHT network can be partially depolymerized into soluble oligomer structures through aminolysis with additional amines.

To complete the recycling loop, the PHT solution in NMP was directly utilized to prepare a fresh PHT aerogel. This was achieved by introducing an equivalent amount of repeating units in PFA, relative to the additional amine functional groups, as a 1.6 wt% solution in NMP. As a result, a stable organogel was first formed and converted into an aerogel by following the standard aerogel preparation steps, including solvent exchange with ethanol and drying with supercritical $CO_2$ (Fig. 2a, b). During the recycling procedure, 100% of the initial chemical resources were utilized in the material preparation, and no purification steps were required. The potential of the ASA recycling process was demonstrated by successfully completing three recycling cycles. The reconstituted aerogels were designated as PHT-A1, PHT-A2, and PHT-A3, corresponding to their respective recycling generation (Supplementary Table S3). The successful formation of PHT for each recycling iteration was confirmed by $^{13}C$ MAS NMR spectroscopy, where the chemical shift around 70 ppm validates the HT structure within PHT-pristine and PHT-As (Fig. 2c)[7]. Furthermore, the results demonstrate the absence of PHT degradation or side reactions during the ASA processes. It is evident that the sole formation of HT structures could be achieved without the presence of unwanted hemiaminal structures, which could lower the thermal stability and mechanical performance of the final materials[7,13]. Regarding the aerogel-specific properties, PHT-As exhibit virtually identical properties compared to PHT-pristine, thus demonstrating great reproducibility (Fig. 2d, e and Supplementary Table S3–7). According to the nitrogen physisorption isotherms, PHT-As exhibit similar hysteresis to PHT-pristine, suggesting comparable mesoporous structures (Supplementary Fig. S7a). This resemblance is also reflected in their nanoscale morphologies, where both PHT-pristine and PHT-As consist of a branch-like skeleton topology (Fig. 2f). Importantly, PHT-As achieve a thermally super-insulating performance with a low thermal conductivity of approximately $19\ mWm^{-1}K^{-1}$. To further evaluate the thermal insulation performance of PHT-As, we employed the Knudsen model to separate the total measured thermal conductivity into gaseous and solid contributions (Supplementary Table S5)[16]. The gaseous conductivity values across different generations remained stable at around $9.0\ mW\ m^{-1}K^{-1}$, while the solid contribution was consistently close to $9.5\ mW\ m^{-1}K^{-1}$. The reproducible solid thermal conductivity can be attributed to the identical chemical composition of PHT-As across the recycling cycles, as confirmed by $^{13}C$ MAS NMR results (Fig. 2c). Likewise, their comparable morphological characteristics, revealed by SEM and BET analyses, account for the consistent gaseous conductivity. In addition, the thermal and mechanical performance of PHT-As are also in line with PHT-pristine, as demonstrated by similar thermal gravimetric analysis (TGA) and uniaxial compression testing results (Supplementary Fig. S7b–c and Supplementary Table S7). Overall, our PHT aerogels show efficient closed-loop recycling under a waste-minimized, resource-conserving, and reproducible fashion by fully utilizing the reactivity of HT moieties.

## Reprogramming of PHT aerogels

The material properties of a polymer are primarily determined by its chemical composition, but tailoring these characteristics after synthesis remains a significant challenge. In contrast, the ASA process offers a unique opportunity for on-demand reprogramming of aerogel features after their initial fabrication. This reprogramming process closely resembles the recycling protocol, with the key distinction being the introduction of different aromatic amines during the ASA process that induce an alteration of material properties (Fig. 3a). To explore this approach, we selected three distinct aromatic amines: 9,9-bis(4-aminophenyl)fluorene (FDA), N,N-bis(4-aminobenzyl)terephthalamide (BAPTPA), and hexa(aminophenyl)-cyclotriphosphazene (HAPP), to modulate the aerogel-specific properties. For example, FDA was used in a NMP solution to partially depolymerize the PHT network by applying 1.5 equivalents of amine functional groups relative to the embedded amine groups. After complete dissolution, an equivalent amount of repeating units in PFA relative to the additional amine functional groups was added as a 1.6 wt% NMP solution, and the mixture was heated to 100 °C for 2.5 h to form an organogel. The resulting organogel was solvent-exchanged to ethanol and dried using supercritical $CO_2$ to yield PHT-B. Following the same procedure, BAPTPA and HAPP were incorporated into PHT aerogels to synthesize PHT-C and PHT-D, respectively (Fig. 3a, b and Supplementary Table S8).

Compared to PHT-pristine, reprogrammed PHT-B exhibits superior characteristics, such as lower bulk density ($< 0.13\ gcm^{-3}$), higher porosity ($> 90\%$), larger specific surface area ($\approx 256\ m^2g^{-1}$), larger mesopore volume ($0.61\ cm^3g^{-1}$), and intrinsic hydrophobicity (Fig. 3c–d, Supplementary Fig. S12a–c, and Supplementary Table S3–6). By calculating the average pore diameter, PHT-B exhibits a much lower average pore diameter (112 nm) compared to PHT-pristine (188 nm, Supplementary Table S4). Its more intricate topology can also be found from SEM, where solid spherical or hemispherical particles embedded within a necklace-like skeleton were observed (Fig. 3e). Notably, PHT-B achieves a significantly lower thermal conductivity ($15.9\ mWm^{-1}K^{-1}$) compared to PHT-pristine, thus improving its thermal insulation performance[14]. To probe this effect, we evaluated both the gaseous and solid conductivity of PHT-B using the Knudsen model (Supplementary Table S5)[16]. PHT-B exhibits markedly reduced gas-phase conductivity, arising from its smaller average pore size that amplifies the Knudsen effect, as well as slightly lower solid-phase conductivity. The pore-size reduction stems from the molecular architecture of FDA, whose spiro-carbon moiety disrupts efficient packing during polymerization, producing a finely structured microporous network[17]. In addition, the incorporation of rigid, non-coplanar spiro units creates extra free volume and a more tortuous pore network. These structural features hinder heat transport within the solid framework, further suppressing heat conduction[18].

As for PHT-C, we selected BAPTPA as the amine feedstock, which was derived from commercial polyethylene terephthalate (PET) bottles[19]. This amide-containing linker can engage in intermolecular hydrogen bonding within the polymer network, thereby enhancing crosslink density and yielding a mechanically more robust structure than PHT-pristine[20]. As anticipated, PHT-C demonstrates a higher compressive modulus (1.86 MPa) and greater compressive strength at a 10% deformation ratio (669 kPa, Supplementary Fig. S12c and Supplementary Table S7). In addition, PHT-C retains its lightweight and highly porous characteristics along with its high specific surface area (Fig. 3d, Supplementary Fig. S12a and Supplementary Table S4). PHT-C also achieves lower thermal conductivity compared to PHT-pristine, making it a more efficient thermally superinsulating material.

Lastly, HAPP was specifically designed to enhance the thermal stability and flame resistance of the aerogel PHT-D. Its structure features hexafunctional aromatic amine moieties that increase crosslink density, while the phosphazene ring serves as an acid and gas source for intumescent flame retardant[5,21]. According to TGA, the fabricated PHT-D shows a similar decomposition temperature at 5% weight loss compared to PHT-pristine and other reprogrammed aerogels, confirming the formation of PHT (Supplementary Fig. S12d and Supplementary Table S7). On top of that, the char yield at 793 °C for PHT-D is found to be around 40%, surpassing other PHT aerogels, which is contributed by high crosslink density and flame retardant phosphazene content. To further investigate the flame resistance of PHT-D, we conducted initial burning tests by exposing PHT-D sample to an open flame for 10 s (Supplementary Fig. S12e **and** Supplementary Movie S1). During the experiment, there was no ignition observed on PHT-D, and

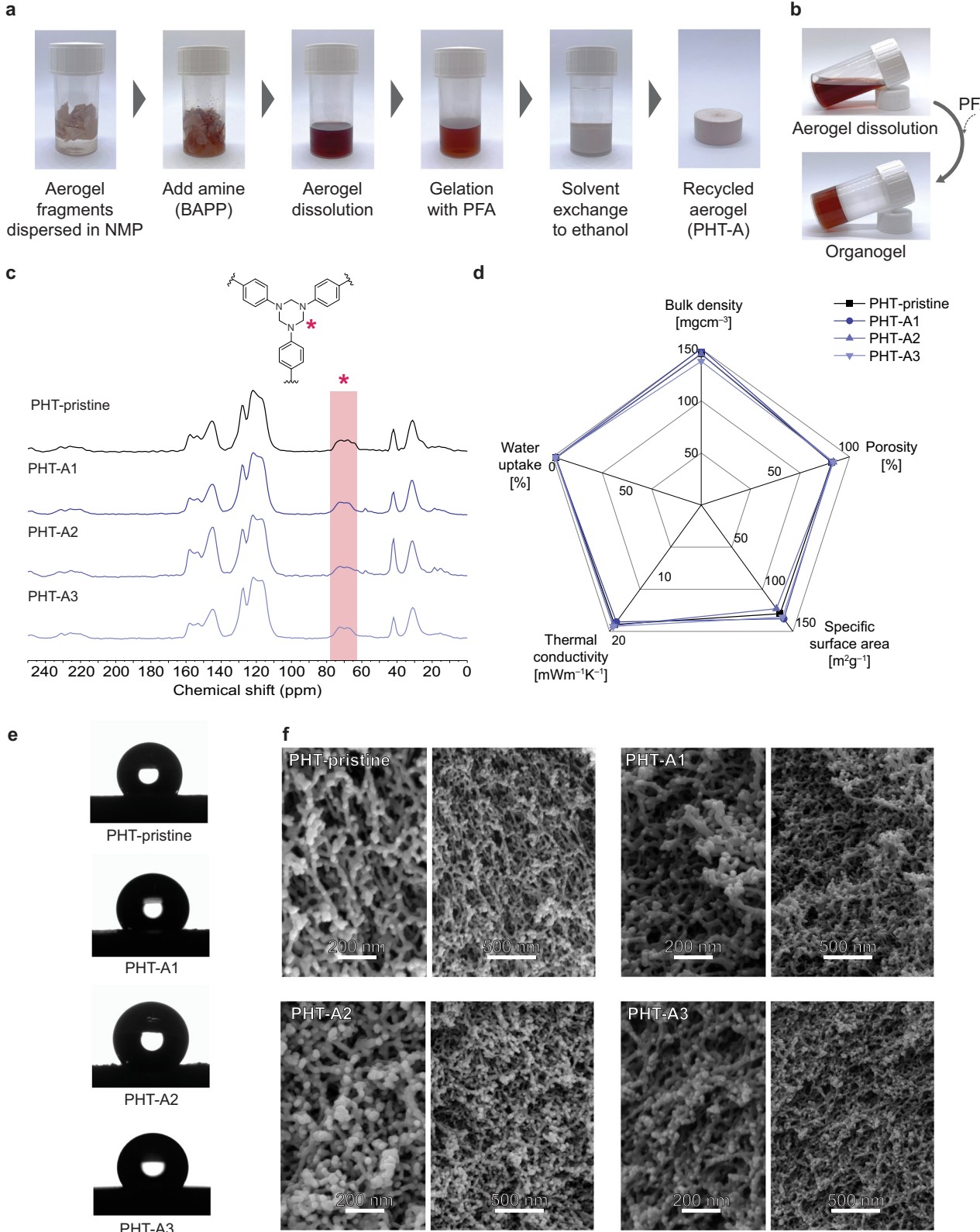

**Fig. 2 | Closed-loop recycling of PHT aerogels. a** Photographs displaying the protocol of closed-loop recycling. PHT-pristine powders were treated with a 14 wt% BAPP solution in NMP, and the mixture was ultrasonicated for 4 h until full dissolution. The equivalent amount of repeating units in PFA was introduced as a 1.6 wt% solution in NMP to initiate the gelation at 100 °C for 4 h. The solvent within the organogel was exchanged to ethanol, and supercritical $CO_2$ drying was applied to produce PHT-A. **b** Schematic illustration of the aerogel dissolution and the organogel prepared from the dissolution. **c** MAS $^{13}$C NMR spectra of PHT-pristine and PHT-As. **d** Radial graph depicting the aerogel-specific properties of PHT-pristine and PHT-As, including density, specific surface area, porosity, water uptake value, and thermal conductivity. **e** Water contact angle images and (**f**) SEM images of PHT-pristine and PHT-As.

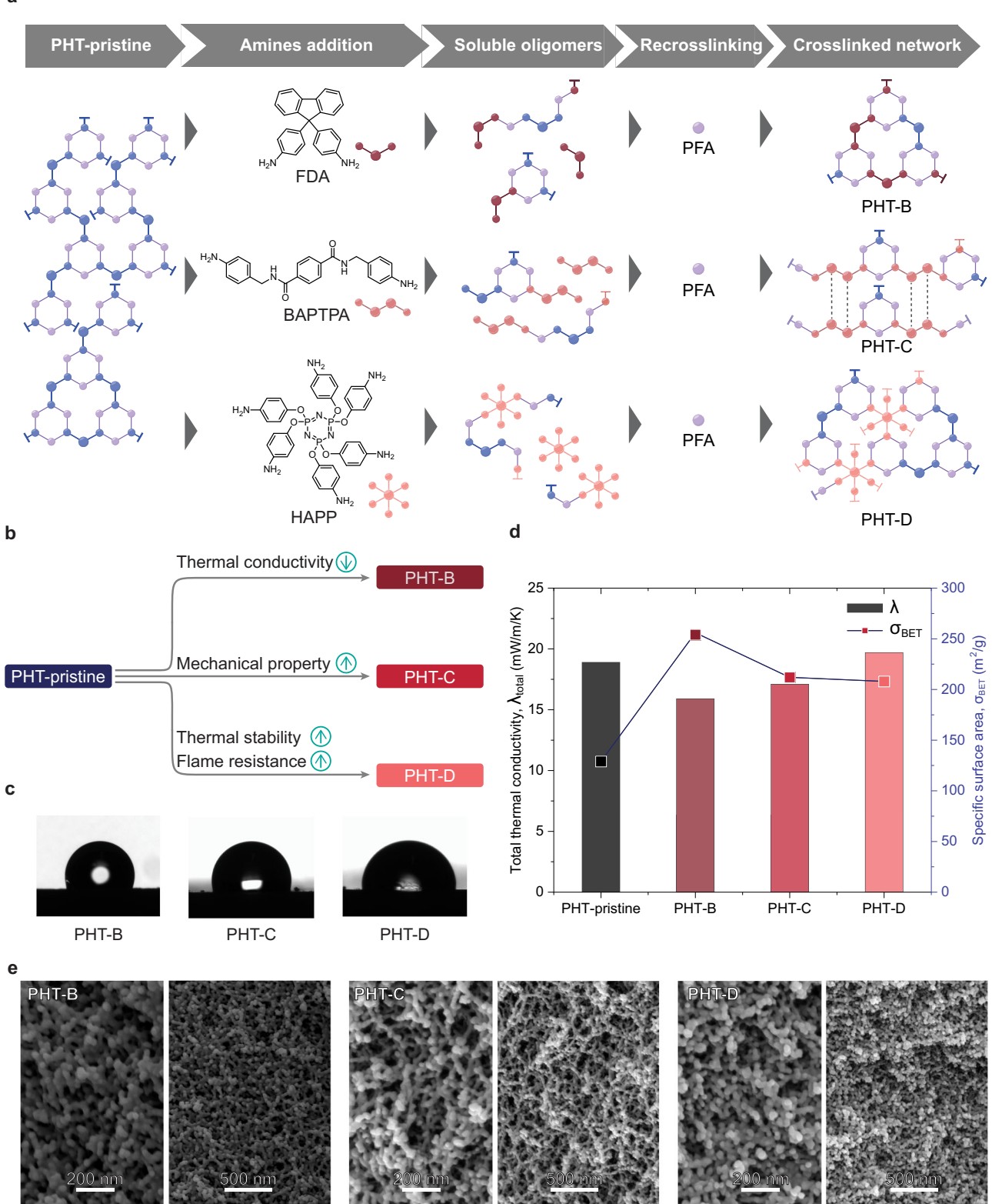

**Fig. 3 | Reprogramming of PHT aerogels. a** Schematic representation of reprogramming PHT-pristine into PHT-B/C/D using different amine feedstocks. **b** Scheme representing the synthesis of PHT-B/C/D from PHT-pristine with the properties intended to improve. **c** Water contact angle images of PHT-B/C/D. **d** Schematic graph comparing the specific surface area and thermal conductivity values of PHT-pristine and PHT-B/C/D. **e** SEM images representing the morphology of PHT-pristine and PHT-Fs.

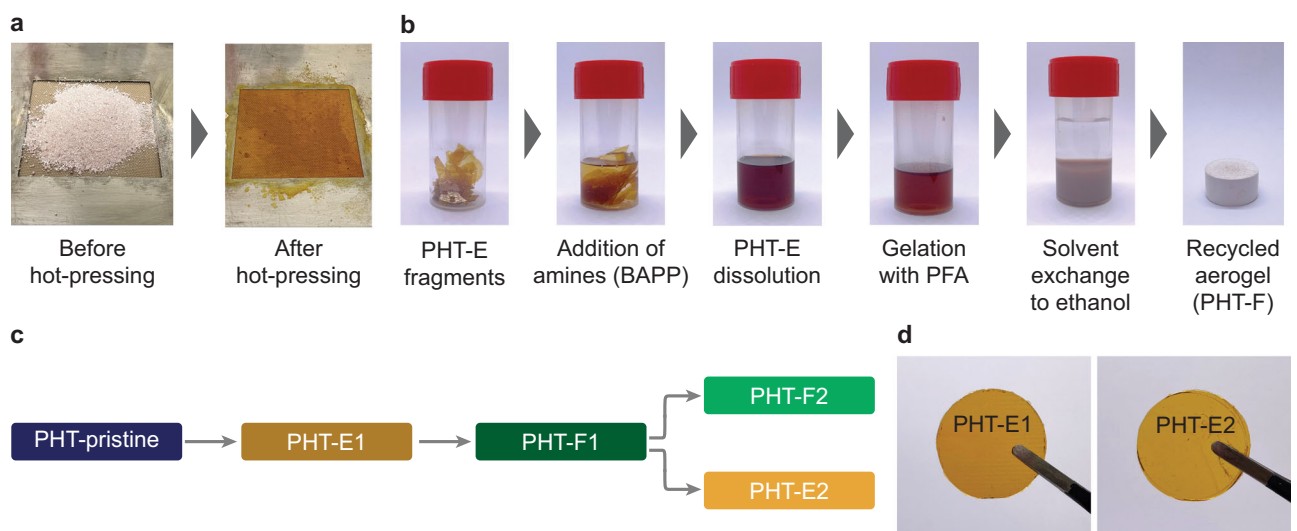

**Fig. 4 | Reprocessing of PHT aerogels. a** Before and after photographs of the preparation of PHT-E from PHT-pristine: PHT-pristine sample was reprocessed via hot-pressing at 180 °C with 12 MPa pressure for 0.5 h followed by 180 °C with 40 MPa pressure for 0.5 h. **b** Photographs displaying the protocol of synthesis of PHT-Fs. Chopped pieces of PHT-E were treated with a 14 wt% BAPP solution in NMP and the mixture was ultrasonicated for 4 h until full dissolution. The equivalent amount of repeating units in PFA was introduced as a 1.6 wt% solution in NMP to initiate the gelation at 100 °C for 4 h. The solvent within the organogel was exchanged to ethanol and supercritical drying with $CO_2$ was applied to produce PHT-F. **c** Reprocessing diagram of PHT-Es (films) and PHT-Fs (aerogels) prepared from PHT-pristine. **d** Photographs of PHT-E1 and PHT-E2.

the combustion of PHT-D stopped almost immediately after removing the sample from the fire. After the burning test, PHT-D shows the formation of a char layer on the surface, which inhibited further burning of its interior (Supplementary Fig. S12f). This validates that the introduction of HAPP enhances both thermal stability and flame resistance. Besides, PHT-D exhibits a higher bulk density and lower porosity than PHT-pristine, which contributes to a higher thermal conductivity value of 19.7 mWm$^{-1}$K$^{-1}$ (Fig. 3d and Supplementary Table S5). While this value is higher compared to other PHT aerogels, PHT-D still outperforms commercially available thermally insulating materials[2,4]. Overall, this reprogramming strategy, enabled by reversible chemistry, marks a significant advancement for organic aerogels in particular, and for the broader realm of reconfigurable materials in general[22,23].

**Reprocessing of PHT aerogels**
To demonstrate the metathesis of HT moieties within the aerogel network, PHT-pristine was reprocessed by compression molding under the same conditions applied in the model reaction. The use of heat and pressure facilitated the reshuffling of the covalent bonds, resulting in the formation of an intact, yellow, transparent thermoset-like polymeric film, labeled PHT-E1 (Fig. 4a). Furthermore, we successfully recycled PHT-E1 following our closed-loop recycling protocol, resulting in the creation of a new aerogel generation, labeled PHT-F1 (Fig. 4b). To demonstrate the versatility and reproducibility of this process, we fabricated another polymeric film (PHT-E2) and a subsequent aerogel generation (PHT-F2) from PHT-F1 using the same procedures (Fig. 4c–d and Supplementary Table S9).

The formation of PHT was determined by $^{13}$C MAS NMR spectroscopy, where the chemical shifts of PHT-E1 and PHT-E2 are in line with those of the PHT-pristine (Supplementary Fig. S13a). It confirms that the chemical structure of PHT is fully retained and no significant side products are formed during reprocessing under heat and pressure. To investigate the mechanical properties of the compact thermoset-like films, tensile testing measurements were conducted (Fig. 5a and Supplementary Table S10). It was observed that both generations of PHT-Es share great mechanical performance with tensile strength of 68 MPa and elongation at break with approximately 3.0 %. PHT-Es also show

high tensile modulus up to 2.2 GPa, in line with reported PHT-based cross-linked polymers[9,24]. Moreover, we performed stress-relaxation experiments using a rheometer to get further insights into the dynamic behavior of HT moieties (Fig. 5b and Supplementary Fig. S13b). The activation energy ($E_a$) for the HT bond exchange reactions was calculated using the Arrhenius equation (Fig. 5c **and** Supplementary Fig. S13c). The values were found to be 312 and 263 kJ/mol for PHT-E1 and PHT-E2, respectively, which are significantly higher than those for other dynamic covalent bonds[25–29].

The thermal properties of PHT-Es and PHT-Fs were investigated using differential scanning calorimetry (DSC), dynamic mechanical thermal analysis (DMTA), and TGA. By DSC measurements, it is found that all the generations of thermoset-like films and aerogels share similar glass transition temperature ($T_g$) in the range of 150–160 °C (Fig. 5d). These values align with the DMTA data, which show that the $T_g$ values of PHT-Es fall between 160–170 °C (Fig. 5e, Supplementary Fig. S13d, and Supplementary Table S11). This consistency suggests that the reprocessing and recycling of thermoset-like films and aerogels were successfully performed without compromising the thermal and mechanical properties of the polymer network. TGA measurements also showed similar decomposition profiles for the aerogel PHT-Fs and film PHT-Es (Fig. 5f). The decomposition temperatures at 5% weight loss indicate the presence of HT structures in the materials, rather than potential hemiaminal byproducts with lower decomposition temperature. To evaluate the resistance of PHT-E1 in various solvent media, solvent stability tests were conducted using a range of protic and aprotic solvents, including chloroform, *n*-hexane, acetone, water, ethanol, methanol, acetonitrile, DMF, and DMSO (Supplementary Fig. S14). After five days, the specimens were weighed, and the mass changes were recorded. All samples demonstrated exceptional stability in the tested solvents. Notably, the highest swelling ratio (228%) was observed in chloroform, indicating strong interactions with the PHT polymer. In addition, the gel fractions of all specimens were calculated to be above 91%, confirming the polymer stability of PHT-E1 (Supplmentary Table S12).

The aerogel-specific properties of the reprocessed aerogels, PHT-F1 and PHT-F2, were investigated using various measurements and compared with PHT-pristine. As shown in the radial graph in Fig. 5g, all

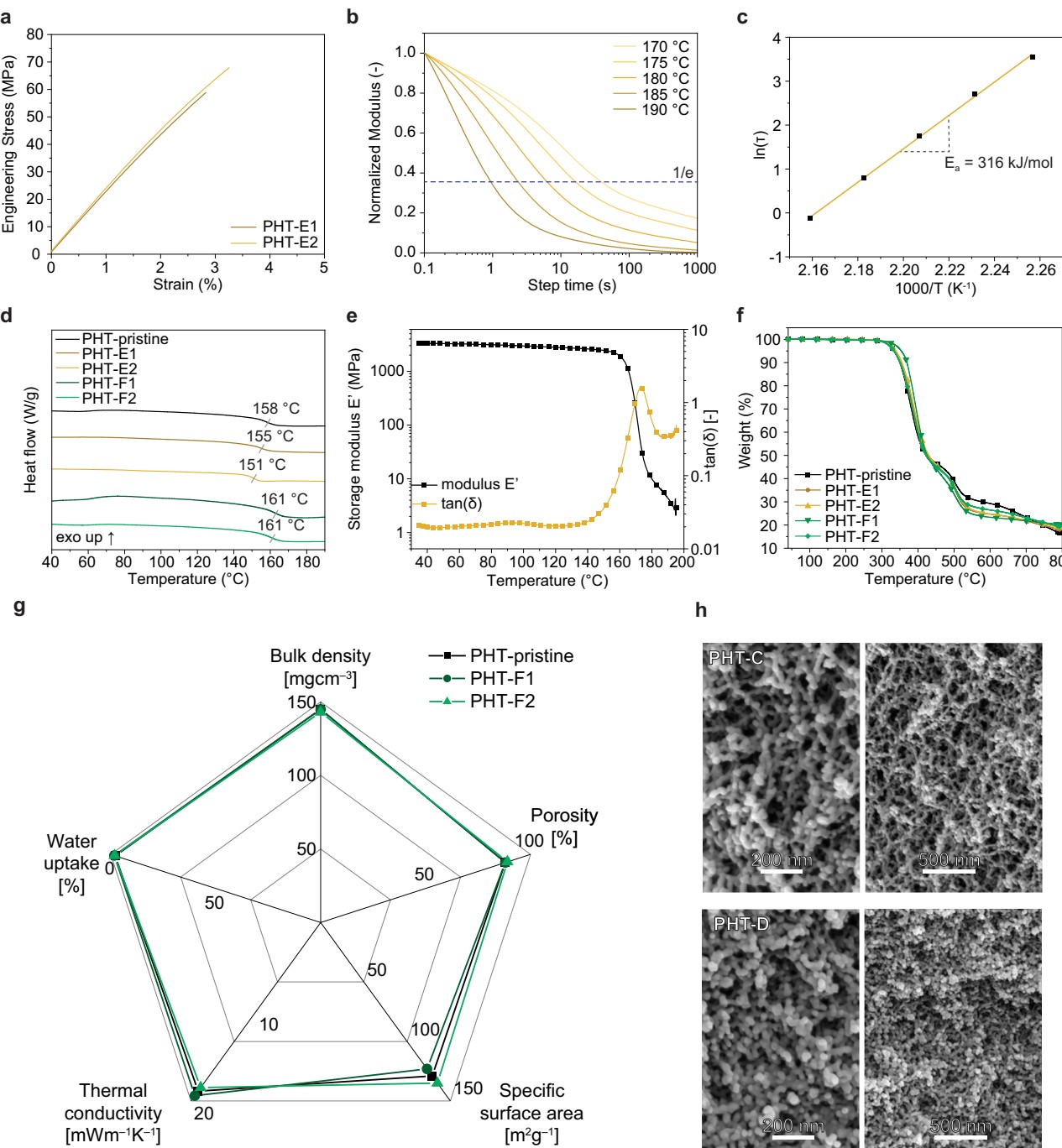

**Fig. 5 | Performance of the reprocessed PHT thermoset-like films and recycled PHT aerogels. a** Tensile testing graph of PHT-E1 and PHT-E2. **b** Normalized stress-relaxation curves of PHT-E1. The stress-relaxation curve of PHT-E2 is provided in the Supporting Information (Supplementary Fig. S13b). **c** Fitted curve for PHT-E1 between 1000/T and characteristic relaxation time (ln τ*) according to the Arrhenius law. **d** DSC measurements of PHT-pristine, PHT-Es and PHT-Fs showing glass transitions. **e** DMTA analysis graph of PHT-E1 showing storage moduli (MPa, black) and tan(δ) (yellow) values. **f** TGA curve of PHT-pristine, PHT-Es, and PHT-Fs ranging from 40 to 793 °C with ramp rate of 10 °C/min. **g** Radial graph depicting the aerogel-specific properties of PHT-pristine and PHT-Fs including bulk density, specific surface area, porosity, water uptake value, and thermal conductivity. **h** SEM images representing the morphology of PHT-F1 and PHT-F2.

aerogel properties, including bulk density, porosity, and specific surface area, show similarity across all aerogel specimens. The water contact angle images also indicate that the intrinsic hydrophobicity is well-preserved throughout the process (Supplementary Fig. S15). More importantly, the thermal conductivity values of PHT-F1 and PHT-F2 align with those of PHT-pristine, achieving the same standard as thermally superinsulating units. PHT-Fs exhibit similar surface area values, which is further reflected in their nanoscale morphology (Fig. 5h and Supplementary Fig. S13e). In addition, the mechanical

properties of PHT-Fs are comparable to PHT-pristine, as indicated by the similar stress-deformation curves (Supplementary Table S13 and Supplementary Fig. S13f). This suggests that the polymer structure of PHT-Fs remains mechanically robust even after undergoing heat and pressure treatment during hot-pressing experiments. Overall, we have successfully demonstrated the reprocessing of these high-performance PHT aerogels into mechanically robust thermoset-like films, which can readily be converted back into thermally super-insulating materials as needed.

## Discussion

In this work, we establish the previously unrecognized capacity of HT structures to engage in aminolysis and metathesis reactions and apply it to achieve waste-minimized foam-to-foam recycling of high-performance PHT aerogels. Using the ASA process, PHT aerogels were successfully recycled over three cycles in a straightforward and highly reproducible manner, while maintaining key material properties, including thermal superinsulation. In addition, by introducing selected amine feedstocks, we demonstrated the ability to reprogram PHT aerogels after their initial fabrication, allowing for tunable material properties and expanding their potential for customizable applications. Beyond aerogels, the reactivity of PHT allowed us to fabricate high-performance thermoset-like films from existing aerogels and vice versa. On top of that, our method eliminates labor-intensive, waste-producing purification steps, incorporating 100% of the original chemical building blocks in the newly formed material. Furthermore, the metathesis reaction of HT moieties provides a versatile pathway for reconfiguring material properties in the solid-state. Overall, this work marks a significant advancement in aerogel recyclability by harnessing the aminolysis and metathesis of PHT and enabling the production of closed-loop recyclable, reprogrammable, and reprocessable aerogel materials. This discovery paves the way for next-generation materials that combine durability with infinite circularity, driving the future of sustainable materials innovation.

## Methods

### Materials

Hexachlorocyclotriphosphazene, 4-acetamidophenol, potassium carbonate, PFA, triazabicyclodecene (TBD), and sodium hydroxide (NaOH) were purchased from Merck life science B.V. *p*-Anisidine and *p*-phenetidine were purchased from TCI Europe B.V. BAPP, FDA, and 4-aminobenzylamine were purchased from BLD Pharmatech Ltd. Ethanol absolute (EtOH), chloroform, isopropanol, acetone, tetrahydrofuran (THF), toluene and DMF were purchased from Biosolve B.V. NMP was purchased from Vivochem B.V. DMSO-$d_6$ (99.9% D) and CDCl$_3$ (99.9% D) were purchased from Cambridge Isotope Laboratories, Inc. Liquid CO$_2$ dip tube grade 2.7, nitrogen grade 5.0, and helium grade 4.6 were purchased by Linde Gas Nederland B.V. Amberlyst 15 hydrogen form was purchased from Sigma Aldrich Ltd and used after washed with distilled water. PET bottle waste is used after being washed with isopropanol.

### Instrumentation

**Nuclear magnetic resonance (NMR) Spectroscopy.** The chemical structures of all chemicals were identified by NMR spectroscopy using a Bruker UltraShield spectrometer (400 MHz for $^1$H NMR, 100 MHz for $^{13}$C NMR, 161.9 MHz for $^{31}$P NMR) at 25 °C using DMSO-$d_6$ or CDCl$_3$ as solvent. All NMR DOSY experiments were performed using a Bruker UltraShield spectrometer (400 MHz) at 25 °C using DMSO-$d_6$ as solvent. Typically, a value of 2 ms was used for the gradient duration (δ), 60 ms for the diffusion time (Δ), and the gradient strength (g) was varied from 0.95 G cm$^{-1}$ to 46.55 G cm$^{-1}$ in 16 steps. Each parameter was chosen to obtain 95% signal attenuation for the slowest diffusion species at the last step experiment. The pulse repetition delay (including acquisition time) between each scan was larger than 2 s. Data acquisition and analysis were performed using MestReNova. The T1/T2 analysis module of Topspin was used to calculate the diffusion coefficients and to create two-dimensional spectra with NMR chemical shifts along one dimension and the calculated diffusion coefficients along the other. The model reactions on the bond exchange reaction of hexahydrotriazine were identified by NMR spectroscopy using a Bruker UltraShield spectrometer (500 MHz for $^1$H NMR, 125 MHz for $^{13}$C NMR) at 25, 65, or 100 °C using DMSO-$d_6$ as solvent.

**Magic angle spinning nuclear magnetic resonance (MAS NMR) spectroscopy.** Solid-state MAS NMR spectra were measured using an 11.7 T Bruker NMR spectrometer operating at 125 MHz for $^{13}$C NMR spectra, respectively. $^{13}$C MAS NMR experiments were performed using a Bruker triple channel 4 mm MAS probe head spinning at 13 kHz. $^{13}$C MAS NMR spectra were recorded using a 13 C{1H} cross polarization (CP) pulse sequence with a ramped contact pulse of 1 ms and an interscan delay of 3 s. NMR chemical shift calibrations of $^1$H and $^{13}$C NMR spectra were done using tetramethylsilane (TMS) and solid adamantane, respectively.

**Matrix-assisted laser desorption/ionization-time of flight mass spectrometry (MALDI-TOF MS).** MALDI-TOF MS of OMeHT and OEtHT were recorded on autoflex® maX, Bruker-MALDI-TOF Mass Spectrometer equipped with a 355 nm Nd:YAG smartbeam laser. α-Cyano-4-hydrocycinnamic acid (CHCA) and 2-[(2E)−3-(4-*tert*-butyl-phenyl)−2-methylprop2-enylidene] malononitrile (DCTB) were used as matrices. The sample was solubilized in THF of chloroform with concentration of 2 mg mL$^{-1}$.

**Gel permeation chromatography (GPC).** GPC measurements of PHT aerogels of dissolution were performed on LC-250C (Shimazu) equipped with two PSS GRAM liner 10 μm columns and a refractive index (RI) detector. The aerogels were firstly dispersed in 0.1 M LiBr/DMF solution at 10 wt%. After ultrasonically treating the solution at 65 °C for 4 h, a clear solution can be obtained. The solution was then diluted into 2 mg/mL solution by 0.1 M LiBr/DMF solution. Later, the solution was filtered with 0.2 μm PTFE filter before analysis. DMF was used as the eluent with a flow rate of 1 mL/min, and the GPC traces were calibrated with a PMMA as a standard to calculate the molecular weight.

**Scanning electron microscopy (SEM).** The morphology of PHT aerogels were characterized by SEM (FEI Quanta 200 3D) at acceleration voltage of 10 kV. The aerogel samples were sputtered with gold for 40 s before testing. PHT aerogels with sample dimensions of 60 mm diameter and 5 mm thickness were used.

**Nitrogen physisorption porosimetry.** The specific surface area and pore size distribution of the aerogels were analyzed by a Brunauer–Emmett–Teller (BET) analyzer (TriStar II Plus). Before measurement, the samples were outgassed at 80 C for 2 h under vacuum conditions. Nitrogen grade 5.0 and Helium grade 4.6 were chosen to measure the physisorption isotherm. PHT aerogels with sample dimensions of 60 mm diameter and 5 mm thickness were used.

**Helium pycnometry.** The porosity and skeletal density of the aerogels were measured by gas pycnometer (AccuPyc II 1345) using Helium grade 4.6. 10 data points were taken with 10 equilibrium cycles. PHT aerogels with sample dimensions of 60 mm diameter and 5 mm thickness were used.

**Thermogravimetric analysis (TGA).** The thermal properties of PHT aerogels were measured by TGA 550 (TA Instruments) under a nitrogen atmosphere at the heating rate of 10 °C/min from 40 to 793 °C. PHT aerogels with sample dimensions of 60 mm diameter and 5 mm thickness were used.

**Differential scanning calorimetry (DSC).** DSC measurements were performed on TA Q2000. 3–5 mg of samples were used in a hermenic Tzero pan. The experiments were carried out from −50 to 200 °C at a rate of 10 °C/min under argon atmosphere. Glass transition temperatures (Tg) were determined by taking the midpoint of the reversible endotherm of the 2nd heating.

**Dynamic mechanical analysis (DMA).** The dynamic mechanical measurements were performed on a TA Instruments DMA850. The experiments were carried out from 40 to 200 °C at a heating rate of 3 °C/min under an oscillatory strain of 0.01% and a frequency of 1 Hz. The glass transition temperature ($T_g$) was recorded as the maximum value of tan δ.

**Stress-relaxation analysis.** Stress-relaxation analysis was performed on a TA Instruments HR20. The relaxation modulus ($G(t)$) was followed over different time periods for a constant applied strain of 1% at the constant temperature (from 190 to 170 °C). The relaxation modulus ($G$) was normalized by the initial value ($G_{0.1s}$).

The activation energy ($E_a$) of the bond exchange reaction was further calculated using the Arrhenius equation:

$$\tau^*(T) = \tau_o \exp(E_a/RT) \qquad (1)$$

where $\tau^*$ is the relaxation time determined via modulus relaxation to $1/e$; $\tau_0$ is the characteristic relaxation time at infinite temperature; $E_a$ is the experimental activation energy (kJ/mol); R is the universal gas constant ($8.314\ \mathrm{J\,K^{-1}\,mol^{-1}}$); and T is the absolute temperature (K).

**Thermal conductivity test.** The thermal conductivity was measured by heat flow meter (Thermtest Inc., HFM-25) at 20 °C and 20 – 30% humidity according to ASTM C518 international standard. PHT aerogels with sample dimensions of 60 mm diameter and 5 mm thickness were used. Prior to the measurement, the machine was calibrated with EPS 1450E as reference material.

**Water uptake test.** Prior to testing, PHT aerogel specimens with 15 mm thickness and 25 mm diameter were placed in 80 °C vacuum oven for 2 h. The mass of a sample was determined before and after submerging it completely under distilled water for 24 h. The water uptake was calculated accordingly in relation to the weight of the sample.

**Contact angle test.** The hydrophobicity of the aerogels was studied by a contact angle analyzer (Data-Physics OCA30) at relative humidity of 40%. PHT aerogels with sample dimensions of 60 mm diameter and 5 mm thickness were used.

**Uniaxial compression test.** Uniaxial compression test was conducted by compression testing machine (ZwickRoell Materials Testing Machine, Zwicki Z2.5/TN). PHT aerogels with sample dimensions of 25 mm diameter and 15 mm thickness were used.

**Solvent resistance measurements.** Solvent resistance and gel content of PHT films were investigated. Prior to the measurements, the specimens were dried in a vacuum oven at 80 °C for 2 h. The samples with 5 mm thickness and 30 mg weight were cut and weighed ($m_o$). The samples were then placed into vials containing different organic solvents, including, chloroform, *n*-hexane, acetone, distilled water, ethanol, methanol, acetonitrile, dimethylformamide, and dimethylsulfoxide. The solutions were kept at room temperature. The appearances of the solutions were recorded by digital camera after 24, 48, and 120 h. After 120 h, the samples were taken out from vials, and the residual solvent on their surface was wiped off before weighing ($m_s$) the samples. Finally, the samples were dried in a vacuum oven at 80 °C overnight, and the mass of the dried samples was weighed ($m_d$). The swelling ratio and gel content were calculated as following equations:

$$\text{Swelling ratio} = (m_s - m_o)/m_o \times 100\% \qquad (2)$$

$$\text{Gel content} = m_d/m_o \times 100\% \qquad (3)$$

## Investigation of bond exchange reaction between OMeHT and *p*-anisidine using ¹H NMR spectroscopy

OMeHT (10.0 mg, 0.025 mmol) and *p*-anisidine (13.7 mg, 0.111 mmol, 4.5 eq.) were dissolved in 1.0 mL of DMSO-$d_6$. The solution was then transferred into an NMR tube and sealed with a septum cap. Subsequently, the NMR tube was placed directly into the NMR machine, and measurements were initiated promptly. ¹H NMR spectra were acquired at 65 °C, with readings taken at 20-minute intervals, employing 32 scans per time point. A total of 13 spectra were taken at a reaction time of 4 h (Supplementary Fig. S1b). After the first reaction was completed, the temperature was raised to 100 °C. A solution of paraformaldehyde (3.35 mg, 0.111 mmol) in 0.1 mL of DMSO-$d_6$ was added to the same NMR tube. The ¹H NMR measurements were started promptly at 100 °C for 4 hours, with readings taken at 20-minute intervals, employing 32 scans per time point (Supplementary Fig. S1c). After 24 h of the end of the reaction, the ¹H and ¹³C NMR measurements were done at 25 °C (Supplementary Fig. S3).

## Pristine polyhexahydrotriazine aerogel synthesis (PHT-pristine)

The PHT organogel was prepared by mixing components A and B. Component A consists of 2.29 g BAPP dissolved in 11.3 g NMP, while component B consists of 0.34 g PFA dissolved in 11.2 g NMP. Both components were prepared in a PP vial by dissolving them in NMP at 100 °C. The gelling was initiated by mixing the two components into one vial at 100 °C. The mixture was shaken until a homogeneous solution was obtained. The solution was poured into a PP vial with 70 mm diameter, and was then placed in an oven at 100 °C until gelation was completed. Afterwards, the organogel was sealed and let aging for 24 h under ambient condition. After aging, the organogel was placed in a solvent bath (300 mL) for solvent exchange. The original solvent was washed out by exchanging the solvent twice, 24 h each time, with 0.1 M sodium hydroxide, distilled water, and ethanol. The ethanol-saturated gel was then transferred to an autoclave, submerged in ethanol, and sealed in a supercritical fluid-extraction autoclave. The pressure was maintained at 100 bar and the temperature was maintained above 60 °C with the constant inflow of $CO_2$. The mixture of solvent and $CO_2$ was vented out multiple times during the drying process while withstanding the pressure and temperature. The aerogel was then stored in a nitrogen oven at 80 °C for 2 h to ensure complete removal of the solvent. The dried sample was stored in a desiccator chamber with relative humidity of 30% to prevent possible moisture uptake.

## Recovery of the solvent used during PHT-pristine synthesis

The solvents used during the solvent exchange step of the PHT-pristine synthesis were collected and processed for recovery. The combined solvent mixture was first subjected to distillation at 80 °C to recover ethanol. Following ethanol separation, the remaining mixture was neutralized using an ion-exchange resin (Amberlyst15) to remove residual sodium ions. After neutralization, the solvent mixture was distilled at 110 °C to sequentially separate water and NMP. The recovered fractions of ethanol, water, and NMP were collected for further characterization, and the recovery yields were calculated (Supplementary Fig. S6 and Supplementary Table S1).

## Investigation of PHT-pristine depolymerization process

PHT-pristine was grinded and kept in the desiccator before use. DMF was selected as the solvent for PHT depolymerization due to its compatibility with MALDI-TOF and GPC measurements, as it does not interfere with mass spectrometry analysis and serves as an effective eluent in GPC. Firstly, 1.05 g of PHT-pristine was added to a vial and a BAPP (1.48 g) solution in DMF (9 g) was added to the aerogel powder. The vial was sealed with parafilm and ultrasonicated at 65 °C for 4 h until full dissolution. For NMR measurement, 0.1 mL of the clear solution was taken out, and 0.4 mL of DMSO-$d_6$ was added to the

solution. The solution was analyzed with several NMR measurements, including $^1H$, $^1H$ COSY, $^{13}C$ DEPT, and $^1H$ DOSY NMR (Supplementary Figs. S9–11).

## PHT aerogel closed-loop recycling

The protocol is divided into two parts: depolymerization and regelation.

**Depolymerization.** PHT-pristine or PHT-A1 were grinded and kept in the desiccator before use. Certain amount of aerogel powders was added to the vial (the detailed formulation is shown in Supplementary Table S3). The PHT aerogel powders were partially depolymerized by adding a 14 wt% BAPP solution in NMP, equivalent to 1.5 times the BAPP content in the aerogel. After the addition, the vial was sealed with parafilm and ultrasonicated for 4 h until full dissolution.

**Re-gelation.** After full dissolution of the mixture is achieved, the solution was placed in 100 °C oven along with PFA dissolved in NMP solution. The gelling was initiated by mixing the two components into one vial. The mixture was shaken until a homogeneous solution was obtained. The solution was poured into either a PP mold with a 70 mm diameter or a PP container with a 26 mm diameter and was placed in the 100 °C oven for 4 h until gelation. Afterwards, the organogel was sealed and aged for 24 h under ambient condition. After aging, the organogel was placed in a solvent bath (300 mL) for solvent exchange. The original solvent was washed out by exchanging the solvent twice, 24 h each time, with 0.1 M sodium hydroxide, distilled water, and ethanol. The ethanol-saturated gel was then transferred to an autoclave, submerged in ethanol, and sealed in a supercritical fluid-extraction autoclave. The pressure was maintained at 100 bar and the temperature was maintained above 60 °C with the constant inflow of $CO_2$. The mixture of solvent and $CO_2$ were vented out multiple times during the drying while withstanding the pressure and temperature. The aerogel was then stored in a nitrogen oven at 80 °C for 24 h to ensure complete removal of the solvent. The dried sample was stored in a desiccator chamber with relative humidity of 30% to prevent possible moisture uptake. The detailed formulation is shown in Supplementary Table S3.

## PHT aerogels reprogramming

The protocol is divided into two parts: depolymerization and re-gelation.

**Depolymerization.** PHT aerogels were grinded and kept in the desiccator before use. Certain amount of aerogel powder was added to the vial (the detailed formulations are shown in Supplementary Table S8). The PHT aerogel powders were partially depolymerized by adding the respective amine solutions in NMP (the molar content of amine functionalities in these solutions is equivalent to 1.5 times the molar content of amine groups of BAPP embedded in the aerogel). After the addition, the vial was sealed with parafilm and ultrasonicated for 4–8 h until full dissolution.

**Re-gelation.** After full dissolution of the mixture was achieved, the solution was placed in 100 °C oven along with the PFA solution in NMP. The gelling was initiated by mixing the two components into one vial. The mixture was shaken until a homogeneous solution was obtained. The solution was poured into either PP mold with a 70 mm diameter or PP container with 26 mm diameter, and was placed in the 100 °C oven for 0.5 to 4 h until gelation. The formed organogel was sealed and let aged for 24 h under ambient condition. After aging, the original solvent was washed out by exchanging the solvent twice, 24 h each time, with 0.1 M sodium hydroxide, distilled water, and ethanol. The ethanol-saturated gels were then transferred to an autoclave, submerged in ethanol and sealed in a supercritical

fluid-extraction autoclave. The pressure was maintained at 100 bar, and the temperature was maintained above 60 °C with the constant inflow of $CO_2$. The mixture of solvent and $CO_2$ were vented out multiple times during the drying while withstanding the pressure and temperature. The aerogel was then stored in a nitrogen oven at 80 °C for 24 h to ensure complete removal of the solvent. The dried sample was stored in a desiccator chamber with a relative humidity of 30% to prevent possible moisture uptake. The detailed formulation is shown in Supplementary Table S8.

## Investigation of bond exchange reaction between OMeHT and OEtHT via hot-pressing experiment

OMeHT (10 mg, 4.06 mmol, 1 eq.) and OEtHT (9 mg, 4.06 mmol, 1 eq.) were mixed and placed between two aluminum plates. The experiment was performed by hot-pressing the aluminum plates with a 12 MPa pressure at 180 °C for 0.5 h, followed by an increasing pressure of 40 MPa at 180 °C for 0.5 h. The light yellow paste was obtained and used for further characterizations.

## PHT aerogels reprocessing

The PHT-pristine or PHT-F1 were chopped into small pieces. Reprocessing was performed by hot-pressing the chopped pieces with a 12 MPa pressure at 180 °C for 0.5 h, followed by increasing pressure of 40 MPa at 180 °C for 0.5 h. A compression mold with dimension of 5 × 5 cm was used. The yielding polymer film (PHT-Es) was used for further characterizations. The protocol of PHT-Fs synthesis was followed by the procedure of PHT aerogel closed-loop recycling. Instead of using PHT-pristine or PHT-A1 as starting materials, chopped pieces of PHT-E1 or PHT-F1 were utilized. The detailed formulation is shown in Supplementary Table S9.

## Data availability

The data that support the findings of this study are available in the supplementary material of this article. All data supporting the findings of this study are available within the paper and its Supplementary Information. All data are available from the corresponding author upon request.

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

## Acknowledgements

We acknowledge the support by the Dutch Ministry of Education, Culture and Science (Gravity Program 024.005.020 – Interactive Polymer Materials IPM). The authors thank Keita Saito and Dr. Özgün Dağlar for laboratory support and scientific discussion. The authors also thank Meng-Han Cheng for the creative input regarding the schematic illustrations.

## Author contributions

C.-L.W., F.E., and Ž.T. conceived the project and designed the experiments. Y.-R.C. performed the model molecule experiments and analyzed the data. C.-L.W. conducted aerogel synthesis experiments and characterizations. C.-L.W. performed reprocessing of film recycling experiments and analyzed the data. B.M. carried out solid-state MAS-NMR measurements. F.E and Ž.T. supervised the project. Ž.T. was responsible for funding acquisition and also directed the research. Y.-R.C., C.-L.W., F.E., and Ž.T. cowrote the paper. All authors read and finalized the paper.

## Competing interests

The authors declare no competing interests.
