## [Transparent Peer Review file · Nature Communications]

Advancing Aerogel Recyclability through Polyhexahydrotriazine Reactivity

Corresponding Author: Professor Zeljko Tomovic

Version 0:

Reviewer comments:

Reviewer #1

(Remarks to the Author)

The manuscript presents a concept of utilizing hexahydrotriazine (HT) structures in recyclable and reprogrammable aerogels and thermosets. However, there are areas where the novelty and scientific impact can be improved. Although the manuscript claims that the HT structure represents a breakthrough in the field of polymer recycling. However, numerous studies have already reported the use of azines as a dynamic covalent functional group. The protocol for preparing dynamic aerogels described in this manuscript may not meet this journal's novelty requirements.

(1) The manuscript only provides qualitative observational data (e.g., NMR spectra) but lacks quantitative evaluation of the dynamic bond exchange process efficiency. It is recommended to introduce more rigorous quantitative analytical methods for assessing the efficiency of dynamic exchange processes. Specific approaches may include: monitoring the depolymerization degree over time, calculating the yield of soluble oligomers, and measuring the repolymerization efficiency to ensure the majority of the original material is retained.

(2) The testing section describes the dynamic exchange reactions but does not sufficiently explore the kinetic characteristics of these reactions. The authors are advised to incorporate a discussion on the kinetics of HT bond exchange reactions, with particular attention to key factors such as exchange rates under varying conditions (e.g., temperature, pressure, or amine concentration). This would contribute to a more in-depth understanding of the reaction speed and its controllability.

(3) The manuscript demonstrates the successful recycling performance of PHT aerogel over two cycles, but it does not address potential performance variations or limitations after multiple cycles. It is recommended to supplement the discussion with test results from a broader range of cycles (e.g., 5–10 cycles) to evaluate the long-term reproducibility and stability of the dynamic exchange reaction.

(4) Although the manuscript hints at the potential applications of HAT chemistry in self-healing materials and vitrimer networks, it lacks sufficient detail or experimental data to substantiate these claims. Additional experimental validation of HAT structures in application scenarios such as self-healing materials and vitreous networks should be provided, or such speculative statements should be removed. If the intention is to propose future research directions, their prospective nature should be explicitly indicated and supported by relevant literature citations.

Reviewer #2

(Remarks to the Author)

My report is attached in MSWord and PDF.

Reviewer #3

(Remarks to the Author)

The manuscript presents an innovative and potentially impactful study on the dynamic covalent behavior of hexahydrotriazine (HT) and its application in fully recyclable aerogels and thermosets. The discovery that HT units can undergo transamination and metathesis reactions under mild conditions opens up new avenues for designing circular polymeric materials. The proposed aerogel-to-sol-to-aerogel (ASA) strategy is compelling and aligns well with the current demand for sustainable materials. However, while the manuscript demonstrates a high level of technical rigor and novelty,

several key aspects require clarification, additional data, and better articulation before it can be considered for publication. I recommend major revision.

1. The manuscript claims unprecedented dynamicity of HT structures, but lacks detailed mechanistic discussion. The authors should provide deeper insights—ideally supported by theoretical calculations (e.g., DFT)—into the activation energy, reaction pathway, and reversibility under realistic conditions.
2. Only two ASA recycling cycles are presented. For a truly circular material platform, a more extensive demonstration (e.g., ≥ 10 cycles) is necessary to validate long-term recyclability and performance retention.
3. Although the authors claim "waste-free" and "100% of the initial chemical resources remained," no quantitative mass balance or recovery yield data are provided. Please include such data to support this strong claim.
4. Although the complex reaction between HTs is considered to be reversible, are there irreversible structures such as secondary condensation and cross-linking densification (e.g., C-C bonding or excessive cross-linking) under hot-pressing conditions, and can these structures be recovered by ASA? Can the authors further confirm this by techniques such as FTIR and XPS?
5. What are the influence patterns of other types of amine substances on the efficiency of ASA and product performance? Can it replace the above-mentioned aniline raw materials?
6. The SEM image only shows the local morphology. It is recommended to add TEM or gas pore size distribution map (such as BJH method) to describe the microstructure of the above sample more comprehensively.
7. NMP is a known hazardous solvent, and its reuse has raised concerns about sustainability. Please discuss potential alternatives or provide reasons for choosing NMP in the context of green chemistry.
8. Many material property data (for example, surface area, thermal conductivity) have no error lines or statistical significance. Please add the standard deviation and repeat the measurement where applicable.
9. Although the thermal conductivity of aerogel is low, does its performance decline in a humid environment? Its long-term stability under humidity, mechanical cycling or ultraviolet radiation should be evaluated. At least the moisture resistance after multiple cycles should be tested and reported.
10. The article points out that the thermal conductivity and mechanical strength can be adjusted by different aromatic amines, but the correlation between the microscopic pore structure and the molecular structure has not been established in detail. Why can the spiro structure of the FDA increase the specific surface area? Can the structural differences of pores at different scales be verified through PALS, BET or SAXS systems?
11. At present, new materials are synthesized only based on different amines, but do different crosslinking degree networks have an inhibitory effect on the reactivity of ASA? A denser network may lead to difficulties in disaggregation.

Version 1:

Reviewer comments:

Reviewer #1

(Remarks to the Author)

The revised manuscript have been much improved and I recommend it for publication in Nature Communications.

Reviewer #2

(Remarks to the Author)

Please see attachment entitled: Leventis Second Review of the Tomovic manuscript.docx

Reviewer #3

(Remarks to the Author)

The revised manuscript can be accepted.

Manuscript number: NCOMMS-25-23496-T

Title: "Advancing Aerogel Recyclability through Undiscovered Polyhexahydrotriazine Reactivity"

Comments to the Editor:

We would like to express our sincere gratitude to you and the reviewers for their time and effort in evaluating our manuscript. We greatly appreciate the insightful comments and suggestions provided. After carefully reviewing the feedback, we have revised the manuscript accordingly to enhance its quality and align it with the expectations of the readership of *Nature Communications*. We have addressed all the reviewers' comments in a point-by-point reply, performed several additional experiments as requested, and made corresponding changes to the manuscript and the Supporting Information. These revisions are highlighted in yellow in the revised manuscript for your convenience.

During the revision process, we also recognized that some terminology in our original submission required clarification. In particular, the terms "dynamic covalent chemistry" and "dynamicity" were not always used with complete accuracy. Since dynamic covalent chemistry applies only to the metathesis reactions of hexahydrotriazine (HT) structures, but not to the irreversible reactions with amines, we now limit the use of this term exclusively to metathesis processes. Accordingly, we have revised the title and relevant sections of the manuscript to avoid the term "dynamicity". Furthermore, we determined that "aminolysis" more accurately describes the reaction with amines than "transamination", and we have updated the manuscript accordingly. These terminology changes are highlighted in blue.

We are confident that the revised manuscript fully addresses the reviewers' concerns and significantly strengthens our work. We believe it will be of great interest to the broad readership of *Nature Communications*.

Point-by-point reply to the reviewers

Reviewer #1 (Remarks to the Author):

The manuscript presents a concept of utilizing hexahydrotriazine (HT) structures in recyclable and reprogrammable aerogels and thermosets. However, there are areas where the novelty and scientific impact can be improved. Although the manuscript claims that the HT structure represents a breakthrough in the field of polymer recycling. However, numerous studies have already reported the use of azines as a dynamic covalent functional group. The protocol for preparing dynamic aerogels described in this manuscript may not meet this journal's novelty requirements.

Reply: We thank Reviewer 1 for critically evaluating our manuscript. We would like to highlight that the unexpected reactivity of polyhexahydrotriazine (PHT) structure in our system represents a substantial advancement in the field. Unlike chemically recyclable thermosets, the PHT-based moiety in our work demonstrates a unique combination of mechanical robustness and recyclability, which has not been achieved previously. This newly discovered reactivity

offers a new pathway for developing high-performance, reprocessable aerogels. We believe this distinction clearly sets our work apart and supports its novelty and scientific impact.

During the revision process, we also recognized that some terminology in our original submission required clarification. In particular, the terms “dynamic covalent chemistry” and “dynamicity” were not always used with complete accuracy. Since dynamic covalent chemistry applies only to the metathesis reactions of hexahydrotriazine (HT) structures, but not to the irreversible reactions with amines, we now limit the use of this term exclusively to metathesis processes. Accordingly, we have revised the title and relevant sections of the manuscript to avoid the term “dynamicity”. Furthermore, we determined that “aminolysis” more accurately describes the reaction with amines than “transamination”, and we have updated the manuscript accordingly. These terminology changes are highlighted in blue. We went carefully through the Reviewer 1’s comments and addressed them point-by-point.

1. The manuscript only provides qualitative observational data (e.g., NMR spectra) but lacks quantitative evaluation of the dynamic bond exchange process efficiency. It is recommended to introduce more rigorous quantitative analytical methods for assessing the efficiency of dynamic exchange processes. Specific approaches may include: monitoring the depolymerization degree over time, calculating the yield of soluble oligomers, and measuring the repolymerization efficiency to ensure the majority of the original material is retained.

Reply: We thank Reviewer 1 for this suggestion. To examine the reactivity of PHT, we first conducted small molecule reactions. According to time dependent ¹H NMR spectroscopy (**Extended data 1**), the aminor structure could be observed after 1 h reaction. Furthermore, these compounds remain stable and intact after 4 h. We also investigated the reversibility of these reaction by adding equimolar amount of repeating units in paraformaldehyde (PFA). We observed that the hexahydrotriazine (HT) structure can be produced without side reactions. We agree with Reviewer 1 that there is a lack of quantitative result of the aminolysis and HT formation process. Thus, we calculated the conversion of reactant (HT) and product (aminol) of small molecule reactions (**Figure R1**). According to the kinetic profile, the aminolysis reached an equilibrium after around 1 h, with 36% aminor formation and 59% hexahydrotriazine (OMeHT) consumption. Notably, depolymerization of PHT aerogels could be accomplished under the same conditions with a longer treatment period (4 h) to reach full dissolution. This could derive from the slower mass transfer of the amine precursor within the highly crosslinked aerogel network. Consequently, we obtained a clear aerogel dissolution without insoluble polymer, which further validates the full depolymerization of PHT aerogels using a primary amine.

Furthermore, we agree with Reviewer 1 that the repolymerization efficiency of soluble aminor oligomers into a PHT network needs more detailed investigation. We calculated the mass balance of all our PHT aerogels, where the mass ratio of the end aerogel product and initial precursors (or reconstituted aerogels) were determined. According to **Table R1**, the mass yields of all PHT aerogels are more than 95%, suggesting that the majority of original precursors and aerogels were fully retained in the new generation samples.

Change: We included **Figure R1** as **Figure S1** and **Table R1** as **Table S14** in the revised Supporting Information. We also revised the manuscript to include the quantitative finding of the model reaction (see page 4, line 10–11).

Figure R1. Kinetic profile of the aminolysis reaction between OMeHT and *p*-anisidine. The time-resolved kinetic NMR experiment (500 MHz, 65 °C, DMSO-*d*₆) was conducted under reaction conditions: OMeHT (1.0 equiv.) and *p*-anisidine (4.5 equiv.) in DMSO-*d*₆ (40 mM) at 65 °C for 4 h with an acquisition interval of every 20 min.

Table R1. Mass balance of PHT aerogels

Nam	Yield (%)
PHT-pristine	95.1 ± 0.5
PHT-A1	96.3 ± 1.8
PHT-A2	95.0 ± 0.3
PHT-A3	94.0 ± 0.4
PHT-B	94.7 ± 5.3
PHT-C	91.6 ± 1.9
PHT-D	96.0 ± 2.4
PHT-F1	97.9 ± 1.1
PHT-F2	95.1 ± 0.5

^{a)}Calculated based on the mass yield of the final aerogels relative to the initial precursors. The calculation was performed using three samples, each with a diameter of 25 mm and a height of 15 mm. The standard deviation was determined from these replicates.

- The testing section describes the dynamic exchange reactions but does not sufficiently explore the kinetic characteristics of these reactions. The authors are advised to incorporate a discussion on the kinetics of HT bond exchange reactions, with particular attention to key factors such as exchange rates under varying conditions (e.g., temperature, pressure, or amine concentration). This would contribute to a more in-depth understanding of the reaction speed and its controllability.

Reply: We thank Reviewer 1 for this insightful suggestion. The reactivity of HT towards primary amines and other HT structures is a newly discovered mechanism that requires more detailed investigation, including reaction rates under varying conditions, theoretical calculations, etc. By harnessing the reactivity of PHT, we aim to provide a novel production pathway of recyclable PHT aerogels that exhibits outstanding material performance. Nonetheless, we are interested in investigating the fundamental base of this unique reaction. We have simultaneously submitted another research article focusing on understanding the

mechanism through experimental and theoretical approaches. As the present manuscript focuses on the sustainable application of this chemical reactivity rather than its mechanistic details, we respectfully refer interested readers to that forthcoming work once it is published.

3. The manuscript demonstrates the successful recycling performance of PHT aerogel over two cycles, but it does not address potential performance variations or limitations after multiple cycles. It is recommended to supplement the discussion with test results from a broader range of cycles (e.g., 5–10 cycles) to evaluate the long-term reproducibility and stability of the dynamic exchange reaction.

Reply: We thank Reviewer 1 for these suggestions. We fully agree that the recyclability and long-term reproducibility of high-performance materials are critical aspects that are worth investigating. In our original manuscript, we demonstrated the recyclability of the PHT aerogels by performing two recycling cycles, which confirmed that the material performance remained consistent.

In response to the Reviewer 1's request, we conducted an additional recycling cycle of PHT aerogels (PHT-A3) from PHT-A2 to further evaluate the reproducibility of its properties (**Figure R2**). The results from this additional cycle, as presented in the revised manuscript (updated **Figure 2, Extended data 4, Table S3–S7**), clearly demonstrate that the thermal insulation performance and related material properties of the PHT aerogels are well maintained, even after multiple recycling processes. In addition, aerogels serve as excellent thermal insulators and are commonly used in construction materials. Such applications typically are utilized with longer service lives that is around 20 years or more.¹ Therefore, we believe that subjecting the PHT aerogels to three recycling cycles is sufficient. Based on the results presented in this manuscript, our PHT aerogels demonstrate excellent durability and reproducibility after multiple recycling generations, which is a core benefit of chemical recycling.

Change: We included the additional aerogel recycling data in the revised manuscript and supporting information (see **Figure 2, Extended data 4, Table S3–S7**). We also revised the manuscript to further discuss the properties of PHT-A3 (see page 6, line 9–11 and line 21–29).

Figure R2. Photographs displaying the protocol of closed-loop recycling of PHT-A2. PHT-A2 powders were treated with a 14 wt% BAPP solution in NMP, and the mixture was ultrasonicated for 4 h until full dissolution. The equivalent amount of repeating units in PFA was introduced as a 1.6 wt% solution in NMP to initiate the gelation at 100 °C for 4 h. The solvent within the organogel was exchanged to ethanol and supercritical CO₂ drying was applied to produce PHT-A3.

4. Although the manuscript hints at the potential applications of HAT chemistry in self-healing materials and vitrimer networks, it lacks sufficient detail or experimental data to substantiate these claims. Additional experimental validation of HAT structures in application scenarios such as self-healing materials and vitreous networks should be provided, or such speculative statements should be removed. If the intention is to propose future research directions, their prospective nature should be explicitly indicated and supported by relevant literature citations.

Reply/change: We thank Reviewer 1 for this suggestion. We agree that the application of PHT chemistry to self-healing or vitrimer materials need further investigation. Although we believe it has strong potential to apply to these applications, we would like to emphasize the application of PHT chemistry on high-performance aerogels and thermosets in this manuscript. Hence, we decided to remove these terms from the manuscript to avoid confusion.

Reviewer #2 (Remarks to the Author):

Comments to Reviewer 2: We thank Reviewer 2 for critically evaluating our manuscript. We went carefully through the comments and addressed them point-by-point.

Point-by-point reply for comments on Introduction

1. The opening sentence - “Organic aerogels, identified by IUPAC as one of the top ten emerging technologies...” - is inaccurate. The relevant IUPAC report refers to aerogels in general, not exclusively organic aerogels. Therefore, it would be more appropriate to revise the opening to: “Aerogels, identified by IUPAC as one of the top ten emerging technologies...”

Reply/change: We thank reviewer for this suggestion. We changed the opening sentence accordingly as follows: “*Aerogels, identified by IUPAC as one of the top ten emerging technologies in chemistry, present a promising material for energy conservation.*” (see page 1, line 26–27).

2. The following sentence, “These materials exhibit ultralow thermal conductivity... 0.020 W m⁻¹ K⁻¹...”, contains an incorrect value. Inorganic aerogels (e.g., silica) have reported thermal conductivities as low as 0.012 W m⁻¹ K⁻¹, and even organic aerogels have achieved values of 0.018 W m⁻¹ K⁻¹ or lower. This figure should be updated for accuracy. Furthermore, a clearer sentence structure would be: “Organic aerogels in particular have shown promise due to their low density and mechanical tunability...”

Reply/change: We thank reviewer for pointing this out. We modified the sentences in the revised manuscript as follows: “*Organic aerogels, in particular, have shown great promise for reducing energy consumption and greenhouse gas emissions due to their ultralow density, great mechanical properties, and exceptional thermal performance.*” (see page 1, line 29–31).

3. Approximately seven lines from the bottom of page 2, the sentence could be rephrased for clarity and impact as: “...the material can be repurposed in every cycle on demand to enhance the thermal conductivity, mechanical strength, or thermal/flame resistance of the next-generation aerogel.”

Reply/change: We thank reviewer for this suggestion. We changed the sentence accordingly as follows: “*By incorporating specific aromatic amines, the material can be repurposed in every*

cycle on demand to enhance the thermal conductivity, mechanical strength, or thermal/flame resistance of the next-generation aerogel.” (see page 2, line 30–32).

4. The statement “These thermosets demonstrate improved thermal and mechanical performance...” lacks a clear point of comparison. Improved relative to what? The original aerogels? Previously reported thermosets? Moreover, as discussed later in conjunction to Part c, the concept of “thermosets” produced via “compression molding” from dynamic covalent polymers should be reconsidered or postponed for a future publication.

Reply/change: We thank Reviewer 2 for this insightful question. The thermosets prepared from PHT aerogels exhibit excellent thermal and mechanical performance with pronounced thermal stability and great mechanical robustness. We have revised the relevant sentence in the manuscript to clarify this point and avoid any confusion (see page 2, line 34–35; page 3, line 1).

Furthermore, we would like to emphasize the significance of thermoset synthesis in our study. Organic aerogels are highly specialized materials known for their outstanding thermal insulation performance, finding use in a wide range of applications. In our work, we demonstrate the circularity of these aerogels through hexahydrotriazine chemistry. Importantly, PHT chemistry can be extended beyond aerogels to other form of polymeric materials. By showing the transformation of PHT aerogels into different forms while maintaining desirable properties, we highlight a crucial capability with significant implications for sustainable materials design. We hope this work will inspire further research into integrating this hexahydrotriazine chemistry into other types of materials, expanding the scope and impact of recyclable and reprocessable thermosets.

5. Figure 1 – Visual and Informational Improvements:

- Font Size: The font in Figure 1 is too small and difficult to read. All labels should be increased in size for legibility.
- Contrast Issues: White font over colored backgrounds suffers from low contrast. Please consider using darker font or adjusting background shades.
- Clarity of Cartoons: Several cartoon illustrations in Figure 1b are too small and low in contrast to be interpreted effectively.
- Missing Label: Please indicate the solvent used (NMP) under the first arrow in Figure 1b.

Reply/change: We thank Reviewer 2 for this suggestion. We have revised the **Figure 1** accordingly.

Point-by-point reply for comments on Part (a) - Dynamicity of the HT Structure

6. Units Correction: The paper states: “A model experiment was conducted ... The process involved applying 30 kN of pressure at 180 °C for 0.5 h, followed by 100 kN of pressure at 180 °C...” This is a misuse of units. Kilonewtons (kN) are units of force, not pressure. The pressure applied should be expressed in units of force per area (e.g., MPa or bar).

Reply/change: We thank Reviewer 2 for this comment. In the hot-pressing experiments, we used a compression mold with dimensions of 5 × 5 cm. Based on our calculations, we applied approximately 12 MPa and 40 MPa of pressure to the mold, respectively, during the process.

We have revised the manuscript accordingly to include these details (see page 4, line 20–21, page 11, 11–13, and page 17, 4–7).

7. Interpretation of Structural Integrity and MALDI-TOF Results:

The authors conclude that the HT structure shows “no signs of degradation” under heat and pressure, as evidenced by consistent ^1H NMR spectra. However, the MALDI-TOF analysis shows that substituent scrambling occurred - an indication of dynamic bond exchange, not static structural integrity. Therefore, the interpretation should be revised. Rather than affirming structural integrity, the data more accurately suggest quantitative dynamic behavior with no apparent degradation or irreversible side reactions - an important and promising result. This should be framed as: “These findings indicate robust dynamicity under thermal and mechanical conditions, comparable to that observed in solution, without evidence of irreversible degradation.”

Reply/change: We thank Reviewer 2 for this comment. We agree with Reviewer 2 that the term “no signs of degradation” could be misleading to readers. We intended to convey that there is no degradation of HT or occurrence of irreversible side reactions, and that only the expected quantitative metathesis reaction takes place. We have revised the manuscript according to the reviewer’s suggestion to clarify this point (see page 4, line 21–23).

Point-by-point reply for comments on Part (b) - Synthesis and Closed-Loop Recycling of PHT Aerogels

8. While the manuscript does not use the term reconstituted aerogels, this terminology is highly accurate and effectively captures the process of iterative recycling while retaining material performance. The authors are encouraged to adopt this term in future revisions.

Reply/change: We thank Reviewer 2 for this suggestion. We changed the term of recycled aerogels into reconstituted aerogels (see page 6, line 10).

9. Terminology Correction (Page 5, Top): The manuscript states: “The PHT aerogel (PHT-pristine) exhibits low bulk density (0.15 g cm^{-3}) and high porosity (88%), key features of lightweighted porous materials.” The term lightweighted is incorrect. It should be replaced with lightweight, which is the proper adjective.

Reply/change: We thank Reviewer 2 for this suggestion. We changed the sentence accordingly (see page 5, line 5).

10. Surface Area and Pore Volume Misinterpretation: The authors report: “PHT-pristine also shows specific surface area of $129\text{ m}^2\text{ g}^{-1}$ and a pore volume of $0.25\text{ cm}^3\text{ g}^{-1}$...” However, based on the reported bulk density ($\rho_b = 0.15\text{ g cm}^{-3}$) and porosity ($\Pi = 88\%$), the skeletal density (ρ_s) is calculated as: $\rho_s = 1.25\text{ g cm}^{-3}$. Using this, the total pore volume (V_{Total}) can be calculated as: $V_{\text{Total}} = (1/\rho_b) - (1/\rho_s) = 5.87\text{ cm}^3\text{ g}^{-1}$. Therefore, the reported pore volume of $0.25\text{ cm}^3\text{ g}^{-1}$ reflects only the mesoporous fraction accessible via nitrogen sorption analysis. The actual pore architecture is predominantly macroporous, as also indicated by the average pore diameter (Φ), calculated using:

$$\Phi = 4 \times V_{\text{Total}} / (\text{specific_suf_area}) = 182\text{ nm}$$

Consequently, the claim of mesoporous architecture is misleading and should be rephrased to reflect the macroporous character of the material.

Reply: We thank Reviewer 2 for this insightful comment. We agree with the Reviewer 2's calculation and acknowledge that the pore volume derived from nitrogen physisorption measurements ($0.25 \text{ cm}^3\text{g}^{-1}$) represents only the mesoporous fraction that is accessible by gas adsorption. As the Reviewer 2 correctly pointed out, the actual total pore volume, based on bulk and skeletal density, is indeed much larger ($6.08 \text{ cm}^3\text{g}^{-1}$ for PHT-pristine), indicating that the majority of the pore structure is macroporous in nature. The calculated average pore diameter (188 nm for PHT-pristine) further supports this interpretation (**Table R2**).

Change: We have revised the manuscript to clarify that the material possesses a predominantly macroporous architecture with a mesoporous fraction as measured by nitrogen sorption. We believe this more accurate description better reflects the true pore structure and avoids potential misinterpretation. We modified the sentence as follows: “*PHT-pristine also shows a large specific surface area of $129 \text{ m}^2\text{g}^{-1}$, a total pore volume of $6.08 \text{ cm}^3\text{g}^{-1}$, and a mesopore volume of $0.25 \text{ cm}^3\text{g}^{-1}$, indicative for its highly porous architecture. It is noted that PHT-pristine possesses a predominantly macroporous architecture with a mesoporous fraction as measured by nitrogen sorption (Extended data 4a).*” (see page 5, line 6–9).

Table R2. General material properties of PHT-pristine

Name	Bulk density ρ_b [mgcm^{-3}]	Linear shrinkage [%] ^{a)}	Skeletal density ρ_s [gcm^{-3}]	Porosity Π [%] ^{b)}	Specific surface area σ_{BET} [m^2g^{-1}] ^{c)}	Total pore volume V_{Total} [cm^3g^{-1}] ^{d)}	Mesopore volume $V_{\text{B,H}}$ [cm^3g^{-1}] ^{e)}	Average pore size Φ [nm] ^{f)}	Total thermal conductivity λ_{Total} [$\text{Wm}^{-1}\text{K}^{-1}$] ^{g)}	Gaseous thermal conductivity λ_g [$\text{Wm}^{-1}\text{K}^{-1}$] ^{h)}	Solid thermal conductivity λ_s [$\text{Wm}^{-1}\text{K}^{-1}$] ⁱ⁾
PHT-pristine	145	17	1.22 ± 0.003	88	129	6.08	0.25	188	$0.0188 \pm 5\text{e-}05$	0.0093	0.0095

^{a)}Linear shrinkage was calculated based on the diameter change of the sample; ^{b)}Porosity was calculated via equation: $\Pi = (1 - \rho_b/\rho_s) \times 100\%$; ^{c)}Calculated based on BET theory. ^{d)}Calculated via $V_{\text{Total}} = (1/\rho_b) - (1/\rho_s)$ ^{e)}Calculated using the BJH method from the desorption branch of the isotherms; ^{f)}Calculated via $\Phi = 4 \times V_{\text{Total}}/\sigma_{\text{BET}}$ specific surface area, σ . ^{g)}Measured with a heat flow meter; ^{h)}Calculated using the Knudsen equation (eqn. 4); ⁱ⁾ Calculated via $\lambda_s = \lambda_{\text{Total}} - \lambda_g$, and assuming that the radiative heat transfer was negligible;

11. Thermal Conductivity Analysis: From the porosity ($\Pi = 0.88$) and the average pore diameter ($\Phi = 182 \text{ nm}$), one can estimate the gaseous thermal conductivity as approximately $9.1 \text{ mW m}^{-1} \text{ K}^{-1}$, down from $\sim 26 \text{ mW m}^{-1} \text{ K}^{-1}$ in open still air, confirming that the Knudsen effect significantly reduces gas-phase conduction. However, (a) as data show the Knudsen effect is not a purely mesoporous effect; and (b) the remaining $\sim 9.9 \text{ mW m}^{-1} \text{ K}^{-1}$ in the total thermal conductivity must be attributed to solid-phase conduction through the polymer framework. This nuance should be emphasized to avoid oversimplifying the origin of the thermal insulation property of aerogels as a purely mesoporous phenomenon.

Reply: We thank Reviewer 2 for this insightful suggestion regarding the interpretation of the thermal conductivity data. We agree that the Knudsen effect, which arises from the mesopore structure, plays a significant role in reducing the gas-phase thermal conductivity. However, as the reviewer correctly points out, the total thermal conductivity of the aerogels must also consider the contribution from solid-phase conduction through the polymer framework.

To clarify this nuance, we calculated the average pore diameter of all PHT aerogels based on total pore volume and specific surface area results (**Table R2** and **Table R3**). Based on the

average pore diameter results, we calculated the gaseous and solid thermal conductivity of PHT aerogels in the following methods:

The total thermal conductivity (λ_{Total}) of all aerogels was partitioned into gaseous (λ_g) and solid (λ_s) conductivities using Knudsen's model. The gaseous thermal conductivity was calculated using the Equation 1:

$$\lambda_g = \frac{\lambda_{g,0} \times \Pi / 100}{1 + 2\beta \frac{l_d}{\Phi}} \quad (1)$$

Where $\lambda_{g,0}$ is the thermal conductivity of still air at 1 atm and room temperature ($0.02619 \text{ W m}^{-1} \text{ K}^{-1}$); Π is the porosity (**Table R2** and **Table R3**); β is the gas-solid energy exchange efficiency factor (assumed to be 2); l_d is the mean free path of nitrogen gas molecules at experimental conditions (taken as 70 nm); Φ is the average pore diameter, as shown in **Table S3**. Radiative heat transfer was assumed negligible. The solid thermal conductivity was obtained by subtracting the gaseous component from the total (**Table R4**), as shown in Equation 2:

$$\lambda_s = \lambda_{\text{Total}} - \lambda_g \quad (2)$$

According to **Table R4**, we found that the PHT-pristine, PHT-As, and PHT-Fs show comparable solid-phase thermal conductivities, as they share the same polymeric backbone and similar network structures. In contrast, PHT-B exhibits slightly lower solid-phase conduction due to the incorporation of spiro carbon structures. Specifically, the rigid, non-coplanar spiro structure disrupts efficient packing during polymerization, generating additional free volume and a more tortuous pore network. This structure could reduce heat transfer within solid framework, causing lower solid heat conduction.² Additionally, PHT-B demonstrates significantly lower gas-phase conductivity because its smaller average pore size enhances the Knudsen effect more effectively. Meanwhile, PHT-C and PHT-D display higher solid-phase thermal conductivities, which can be attributed to their higher bulk density and solid fractions, resulting in more continuous heat transfer pathways through the polymer network.

Change: We have revised the manuscript to emphasize that the excellent thermal insulation performance of these PHT aerogels arises from both the gas-phase conduction reduction via the Knudsen effect and the tailored solid-phase thermal conductivity, which depends on the network structure and composition (see page 5, line 9–14; page 6, line 24–29). We also included **Table R3** and **Table R4** along with calculation methods in the revised Supporting Information (**Table S4** and **Table S5**).

Table R3. General material properties of PHT aerogels

Name	Bulk density ρ_b [mgcm ⁻³]	Linear shrinkage [%] ^{a)}	Skeletal density ρ_s [gcm ⁻³]	Porosity Π [%] ^{b)}	Specific surface area σ_{BET} [m ² g ⁻¹] ^{c)}	Total pore volume V_{total} [cm ³ g ⁻¹] ^{d)}	Mesopore volume V_{BJH} [cm ³ g ⁻¹] ^{e)}	Average pore size Φ [nm] ^{f)}
PHT-A1	150	16	1.28 ± 0.007	88	135	5.89	0.21	174
PHT-A2	145	18	1.31 ± 0.007	89	123	6.13	0.21	199
PHT-A3	138	15	1.34 ± 0.007	89	133 ± 1.19 ^{g)}	6.50	0.37	195
PHT-B	126	12	1.33 ± 0.008	91	256	7.18	0.61	112
PHR-C	143	18	1.32 ± 0.002	85	212	6.24	0.57	118
PHT-D	196	26	1.35 ± 0.004	84	208	4.36	0.78	84
PHT-F1	145	14	1.22 ± 0.004	88	123	6.08	0.35	198
PHT-F2	143	13	1.27 ± 0.010	89	137	6.21	0.35	181

^{a)}Linear shrinkage was calculated based on the diameter change of the sample; ^{b)}Porosity was calculated via equation: $\Pi = (1 - \rho_b / \rho_s) \times 100\%$; ^{c)}Calculated based on BET theory. ^{d)}Calculated via $V_{\text{Total}} = (1/\rho_b) - (1/\rho_s)$ ^{e)}Calculated using the BJH method from the desorption branch of the isotherms; ^{f)}Calculated via $\Phi = 4 \times V_{\text{Total}} / \text{BET specific surface area, } \sigma$. ^{g)}The measurements were repeated two additional times, and the average value and standard deviation were determined.

Table R4. Thermal conductivities of PHT aerogels

Name	Total thermal conductivity λ_{total} [Wm ⁻¹ K ⁻¹]	Gaseous thermal conductivity λ_g [Wm ⁻¹ K ⁻¹]	Solid thermal conductivity λ_s [Wm ⁻¹ K ⁻¹]
PHT-pristine	0.0188 ± 5e-05	0.0093	0.0095
PHT-A1	0.0185 ± 4e-05	0.0088	0.0097
PHT-A2	0.0191 ± 7e-05	0.0097	0.0094
PHT-A3	0.0192 ± 1e-03	0.0096	0.0096
PHT-B	0.0159 ± 2e-05	0.0068	0.0091
PHR-C	0.0171 ± 5e-06	0.0066	0.0105
PHT-D	0.0197 ± 3e-05	0.0051	0.0146
PHT-F1	0.0194 ± 1e-04	0.0095	0.0099
PHT-F2	0.0185 ± 5e-05	0.0092	0.0093

^{a)}Measured with a heat flow meter; ^{b)} Calculated using the Knudsen equation (eqn. 1); ^{c)} Calculated via $\lambda_s = \lambda_{\text{Total}} - \lambda_g$, and assuming that the radiative heat transfer was negligible;

12. Figure Recommendations:

- SEM Imaging: All SEMs presented in Figures 2 and 3 would benefit from dual magnification, with one half at low magnification for overall morphology and the other at high magnification to show fine features (see for example N. Leventis' publications for reference).
- Figure Formatting: Please apply the same feedback regarding font size, contrast, and label clarity from Figure 1 to Figures 2 through 5. Specifically, all text should be legible without magnification, and white text on colored backgrounds should be adjusted for improved readability.

Reply/change: We thank Reviewer 2 for this suggestion. We changed the SEM images in **Fig. 2f**, **Fig. 3e** and **Fig. 5h** according to Reviewer's request. We also improve the readability of all the figures regarding font size, contrast, and clarity.

13. Before closing by discussion of Part b, I feel compelled to comment that a particularly remarkable point from this section is that well-defined monomers are only used in the initial synthesis. In subsequent cycles, following disintegration of the first-generation aerogel, the reactive feedstock becomes a solution of oligomers. Despite the lack of defined monomer units, or precise knowledge about the chemical identity of the oligomers, this mixture behaves functionally as if it were composed of pure monomers, yielding second- and third-generation aerogels that are practically indistinguishable from the original in terms of physical properties.

This observation motivates a more precise use of terminology: whenever possible, disintegration is preferable to depolymerization, as the latter may imply full reversion to monomeric species, which is not the case here. This distinction is important for accurately describing the chemical process at work.

Reply/change: We thank Reviewer 2 for this insightful evaluation on our work. We agree that the term *depolymerization* typically implies the dissociation of a polymer into its corresponding monomeric species, which does not fully convey our ASA process. However, according to the IUPAC definition, *disintegration* refers to fragmentation into particles of a defined size, which does not fully capture the process presented in our manuscript either.³ In our case, the aminolysis reaction yields a mixture of monomeric and oligomeric species, rather than well-defined oligomer units. For this reason, we believe it is more accurate to continue referring to the process as *depolymerization*, though with the clarification that it is *partial depolymerization* or dissolution of the cross-linked network. We revised our manuscript accordingly with the new terms.

Point-by-point reply for comments on Part (c) - Reprogramming of PHT Aerogels.

Evaluation and Suggestions:

This section is both innovative and well-supported. It demonstrates how a single dynamic polymer network architecture can be reprogrammed on demand to meet diverse functional requirements—a hallmark of next-generation sustainable materials. That said, several opportunities for enhancement exist:

14. Comparative Data Presentation: A similar analysis of the total thermal conductivity to that presented in Part b in terms of the gaseous and solid contributions would be helpful in assessing the effect of the molecular rigidity in the case of the rigid diamine with a spiro-

carbon moiety. It will be very instructive to see how much of the lowering in the thermal conductivity ($15.9 \text{ mW m}^{-1} \text{ K}^{-1}$ vs. $19 \text{ mW m}^{-1} \text{ K}^{-1}$) can be attributed to the rigidization of the skeletal framework brought about by the spiro structure.

Reply: We thank Reviewer 2 for this insightful comment. As addressed in our response to question 11, we have calculated the average pore diameter of PHT-B, which enables us to further determine both its gas-phase and solid-phase thermal conductivities (see **Table R4**).

According to these results, PHT-B exhibits a slightly lower solid-phase thermal conductivity compared to PHT-pristine. This can be attributed to the incorporation of spiro carbon structures, where this rigid, non-coplanar spiro structure disrupts efficient packing during polymerization, generating additional free volume and a more tortuous pore network. This structure could reduce heat transfer within solid framework, causing lower solid heat conduction.²

In addition, PHT-B shows a significantly reduced gas-phase thermal conductivity. This is primarily due to its smaller average pore size, which enhances the Knudsen effect. To investigate this, we have included the pore size distribution profiles of PHT-B and PHT-pristine (**Figure R3**). As shown in **Figure R3**, PHT-B possesses a higher pore volume in the 10–50 nm range compared to PHT-pristine, further supporting the more pronounced Knudsen effect and consequent reduction in gas-phase thermal conductivity.

Change: We added a detailed discussion of the respective contributions of the gaseous and solid phases to the overall thermal conductivity of PHT-B in the revised manuscript (page 8, line 28–35; page 9 line 1–2 and **Table S5**). We also included **Figure R3** In the revised manuscript as **Extended data 5b**.

Figure R3. Pore size distribution of PHT-pristine, PHT-B, PHT-C, and PHT-D.

15. Mechanistic Insights: A more detailed discussion of how each structural feature (e.g., spirocenters, amide linkages, phosphazine rings) affects the network architecture and physical properties would strengthen the section’s depth.

Reply/change: We thank Reviewer 2 for this suggestion. In the revised manuscript, we expanded the following section to give more details on how the spiro carbon moiety in the FDA unit affects the network architecture and thermal transport properties.

“Notably, PHT-B achieves a significantly lower thermal conductivity ($15.9 \text{ mWm}^{-1}\text{K}^{-1}$) compared to PHT-pristine, thus improving its thermal insulation performance. To probe this effect, we evaluated both the gaseous and solid conductivity of PHT-B using the Knudsen model (Table S5). PHT-B exhibits markedly reduced gas-phase conductivity, arising from its smaller average pore size that amplifies the Knudsen effect, as well as slightly lower solid-phase conductivity. The pore-size reduction stems from the molecular architecture of FDA, whose spiro-carbon moiety disrupts efficient packing during polymerization, producing a finely structured microporous network. In addition, the incorporation of rigid, non-coplanar spiro units creates extra free volume and a more tortuous pore network. These structural features hinder heat transport within the solid framework, further suppressing heat conduction.” (see page 8, line 28–35 and page 9, line 1–2).

For PHT-C, we added the following sentence to further explain the influence of amide moiety with aerogel network on its mechanical performance: *“This amide-containing linker can engage in intermolecular hydrogen bonding within the polymer network, thereby enhancing crosslink density and yielding a mechanically more robust structure than PHT-pristine.”* (see page 9, line 4–6).

As for PHT-D, the section on page 9, line 12–13, We have described how the hexafunctional aromatic amine moieties and the phosphazene ring structure contribute to higher crosslink density, char formation, and flame resistance, thereby improving thermal stability: *“Its structure features hexafunctional aromatic amine moieties that increase crosslink density, while the phosphazene ring serves as an acid and gas source for intumescent flame-retardant.”* We decided to keep these sentences as it clearly states the effect of phosphazene ring structure on PHT aerogel.

16. Broader Context: The reprogramming concept would benefit from a brief connection to related work in the dynamic covalent chemistry literature - particularly in the context of adaptive materials.

Reply/Change: We thank Reviewer 2 for this suggestion. We agree that including additional references related to reprogrammable materials will strengthen the context and further highlight our reprogramming process. Accordingly, we have incorporated relevant literatures in the revised manuscript (see page 9 line 26–28).

Point-by-point reply for comments on Part (d) - Reprocessing of PHT Aerogels.

Critical Evaluation: Conceptual Concerns with Part (d):

While this section is mostly technically sound, I have reservations about its contribution to the overall narrative of the paper. Up to this point, Parts (a) through (c) build momentum through increasingly impactful demonstrations of the dynamic capabilities of the HT/PHT platform. In contrast, Part (d) feels like a conceptual regression - a shift into lower gear, if you wish. Since hot-pressing does not degrade the HT system (as shown by the control experiments in Part a), the solution-phase chemistry of the hot-pressed aerogels is, as expected, identical to that of the original PHT materials. Thus, it could be argued that there is little new insight gained from this section. If the intent was to demonstrate that PHT aerogels can be reconfigured as dense plastics, that is valid, but not particularly unique to the PHT aerogels or relevant to this manuscript.

The most serious, in my opinion, issue lies in the characterization of the hot-pressed films as “thermosets.” Based on both experimental data and standard polymer definitions, this classification is not justified:

- True thermosets are irreversibly cured materials that cannot be reshaped or reprocessed. This typically results from heat-induced crosslinking that forms a permanent, tightly bonded network.
- By contrast, the PHT-derived films are chemically identical to the pre-hot-pressed samples, and therefore can be dissolved in the presence of primary amines and re-gelled into aerogels that are chemically and mechanically indistinguishable from the pristine material.
- The reported insolubility of the hot-pressed films is not caused by hot-press-induced crosslinking. Rather, it is an inherent property of the pre-existing PHT network, which, by its nature, is already crosslinked. There is no evidence of new, irreversible curing during hot pressing.

In this context, the repeated assertion that the polymer structure remains unchanged after heat and pressure - while highlighting the robustness of the network - also undermines the novelty of the section. The chemical behavior is unchanged, and therefore the added value of this part is limited.

Alternative and More Impactful Experimental Directions:

Rather than revisiting dynamics already demonstrated in solution (Part b), this section could have explored a solid-state reprogramming strategy, which would better align with the transformative scope of the paper - and with the model hot-pressing experiments from Part (a):

- Scenario 1: Hot-press a pristine aerogel in the presence of a different diamine to trigger amine exchange under heat and pressure. This could enable solid-state reconfiguration of the polymer network and aerogel properties.
- Scenario 2: Extend this by hot-pressing the aerogel with both a new diamine and paraformaldehyde, attempting a full solid-state network reconstruction. Though ambitious (read: a long shot), this could eliminate solvent use entirely and further enhance the sustainability profile of the system. Either approach would provide an opportunity to expand the principles established earlier into a new application phase, adding originality and technical depth to the manuscript

Reply: We thank Reviewer 2 for this insightful comments. We understand the reviewer’s concern that the hot-pressing demonstration may appear to reiterate rather than advance the dynamic system capabilities established in previous sections. Our goal in including this section was to illustrate the practical versatility of the PHT chemistry, specifically, its ability to be reprocessed from a highly porous aerogel into a dense bulk material. Furthermore, the hot-pressing experiment demonstrates the bond-exchange reconfiguration solely under heat and pressure, without the usage of solvent as introduced in our ASA process.

More importantly, the reprocessed thermoset presented in this work possesses inherent recyclability, enabling its transformation into next-generation high-performance aerogels,

which is the primary focus of this research. We agree with the reviewer's definition of a true thermoset as an irreversibly cured material that cannot be reshaped or reprocessed. The PHT network, by contrast, is inherently reversible under appropriate conditions (e.g., in the presence of primary amines) that can be transformed back to organic aerogels. The PHT thermosets still possess strong mechanical/thermal properties as conventional thermosets. In this light, we decided to keep the term of thermosets in the manuscript, which we believe appropriately conveys our material.

Regarding the alternative experimental directions, we appreciate the reviewer's thoughtful suggestions and agree that, in principle, solid-state reprogramming could be an interesting extension of PHT chemistry.

However, we believe that these proposed scenarios are outside the conceptual framework of this study. Our primary focus is to demonstrate the versatile reactivity of PHT moieties on high-performance organic aerogels, and the possibility to revert them to strong polymeric thermoset. In our view, introducing solid-state reprogramming experiments would shift the emphasis away from the main objectives of the current work. Nonetheless, we acknowledge the reviewer's perspective and added a brief statement regarding this possibility in the discussion section. At the same time, we are convinced that the experiments presented here provide a clear and coherent demonstration of the key principles.

Change: We added the following sentences in the discussion section for future study: *“Furthermore, the metathesis reaction of HT moieties provides a versatile pathway for reconfiguring material properties in the solid-state. Overall, this work marks a significant advancement in aerogel recyclability by harnessing the aminolysis and metathesis of PHT and enabling the production of closed-loop recyclable, reprogrammable, and reprocessable aerogel materials.”* (see page 14, line 11–14).

Point-by-point reply for comments on Conclusions

The Conclusions section effectively summarizes the key innovations of the study. However, the statement: “Beyond aerogels, the dynamic characteristics of PHT allowed us to fabricate high-performance thermosets from existing aerogels and vice versa” is misaligned with the evidence provided. Given that the so-called “thermosets” remain fully reprocessable, chemically identical to the pristine samples, and lack irreversible curing, this statement overreaches and risks diminishing the technical rigor of the conclusions drawn from Parts (a)–(c).

I support publication pending revisions that:

- Correct or remove the "thermoset" terminology from the Title of the article and claims in Part (d);
- Consider replacing Part (d) with a more forward-looking experiment; and
- Adjust the Conclusions to accurately reflect the scope and novelty of the demonstrated results.

Reply: We thank Reviewer 2 for this comment. As noted in our previous responses, the reprocessed material presented in this work demonstrates the versatility of the dynamic PHT system, which is capable of transitioning from a porous aerogel to a dense bulk form and then be reverted back to its porous state. This reversible reconfiguration leverages the characteristics of this dynamic system. In this context, we also agree with Reviewer 2 that the term “*thermoset*”

in the original title may not fully reflect the central scope of the manuscript. Accordingly, we have removed the term “thermoset” from the title in the revised version. In addition, we have also revised the conclusion, which more accurately reflects the demonstrated results (see page 14, line 2–4). To acknowledge Reviewer 2’s valuable input, we have added a discussion of the proposed alternative experimental directions in the Conclusion section to reflect the potential for future research building upon these results (see page 14, line 11–12).

Reviewer #3 (Remarks to the Author):

The manuscript presents an innovative and potentially impactful study on the dynamic covalent behavior of hexahydrotriazine (HT) and its application in fully recyclable aerogels and thermosets. The discovery that HT units can undergo transamination and metathesis reactions under mild conditions opens up new avenues for designing circular polymeric materials. The proposed aerogel-to-sol-to-aerogel (ASA) strategy is compelling and aligns well with the current demand for sustainable materials. However, while the manuscript demonstrates a high level of technical rigor and novelty, several key aspects require clarification, additional data, and better articulation before it can be considered for publication. I recommend major revision.

Reply: We are thankful to Reviewer 3 for critically evaluating our manuscript and giving us constructive feedback. We addressed the comments carefully and answered them point-by-point.

1. The manuscript claims unprecedented dynamicity of HT structures, but lacks detailed mechanistic discussion. The authors should provide deeper insights—ideally supported by theoretical calculations (e.g., DFT)—into the activation energy, reaction pathway, and reversibility under realistic conditions.

Reply: We fully recognize the importance of understanding the fundamental aspects of this unique dynamic reaction. We have submitted a separate research article dedicated to exploring the reaction mechanism in detail through both experimental work and DFT calculations. As these discussions would be out of scope for this study, we respectfully refer to this article once published.

2. Only two ASA recycling cycles are presented. For a truly circular material platform, a more extensive demonstration (e.g., ≥ 10 cycles) is necessary to validate long-term recyclability and performance retention.

Reply: We thank Reviewer 3 for the suggestion. We fully agree that the recyclability and long-term reproducibility of high-performance materials are critical aspects that are worth investigating. In response to the Reviewer 3’s request, we conducted an additional recycling cycle of PHT aerogels (PHT-A3) from PHT-A2 to further evaluate the reproducibility of its properties (**Figure R2**). The results from this additional cycle, as presented in the revised manuscript (updated **Figure 2, Extended data 4, Table S3–S7**), clearly demonstrate that the thermal insulation performance and related material properties of the PHT aerogels are well maintained, even after multiple recycling processes. In addition, aerogels serve as excellent thermal insulators and are commonly used in construction materials. Such applications typically are utilized with longer service lives that is around 20 years or more.¹ Therefore, we believe that subjecting the PHT aerogels to three recycling cycles is sufficient. Based on the results presented in this manuscript, our PHT aerogels demonstrate excellent durability and reproducibility after multiple recycling generations.

Change: We included the additional aerogel recycling data in the revised manuscript and supporting information (see **Figure 2**, **Extended data 4**, **Table S3–S7**). We also revised the manuscript to further discuss the properties of PHT-A3 (see page 6, line 9–11 and line 21–24).

3. Although the authors claim "waste-free" and "100% of the initial chemical resources remained," no quantitative mass balance or recovery yield data are provided. Please include such data to support this strong claim.

Reply: We thank Reviewer 3 for this suggestion. We initially described our aerogel recycling process as "waste-free" to emphasize that all chemical components can be recycled or recovered for reuse. In response to this point, we carefully collected all solvents used during the gelation, solvent exchange, and supercritical drying steps. Ethanol was recovered via distillation with recovery yield of 96%. The sodium ions in the residue mixture were removed using ion exchange resins. Notably, we recovered up to 62% of the NMP and 93% of the water used, both with high purity (**Figure R4** and **Table R5**). While we acknowledge that these recovery yields are not yet optimal, which resulted from small scale experiments performed in the lab (9 g NMP in 400 mL ethanol/H₂O mixture). We believe they can be significantly improved at larger production scales. As shown in **Figure R4**, the recovered solvents exhibit high purity, with no detectable side products or residual monomers, demonstrating their suitability for reuse in subsequent aerogel synthesis cycles.

In addition, we have calculated the overall mass balance of our PHT aerogels. Specifically, we determined the mass ratio of the final aerogel product relative to the initial precursors (or regenerated aerogels). As summarized in **Table R1**, the mass yields of all PHT aerogels exceed 95%, indicating that the majority of the original components are effectively retained in the recycled samples.

We agree with Reviewer 3 that the terms "waste-free" and "100% of the initial chemical resources remained" are strong claims, and that our current quantitative data do not fully support them. Accordingly, we have revised the manuscript to replace these terms with "waste-minimized" and "100% of the initial chemical resources are utilized (incorporated)," which we believe more accurately reflect the scope and results of our process.

Change: We replaced the terminologies of "waste-free" and "100% of the initial chemical resources remained" with "waste-minimized", and "100% of the initial chemical resources are utilized" respectively in the revised manuscript. We also revised the manuscript to include a detailed description of the recovery of all solvents used during the PHT-pristine synthesis (see page 5, line 2–4). We also added details of the solvent recovery process and its results in the Supporting Information. Additionally, we have added **Table R1** as **Table S1** in the supporting information, which presents the mass balance of the PHT aerogel synthesis, supporting the closed-loop nature of our approach.

Figure R4. ^1H NMR spectra of recycled ethanol, recycled water, and recycled NMP (400 MHz, 25 °C, $\text{DMSO-}d_6$).

Table R5. Recovery yield of the solvents

Components	Yield [%]
Ethanol	96
Water	93
NMP	62

4. Although the complex reaction between HTs is considered to be reversible, are there irreversible structures such as secondary condensation and cross-linking densification (e.g., C-C bonding or excessive cross-linking) under hot-pressing conditions, and can these structures be recovered by ASA? Can the authors further confirm this by techniques such as FTIR and XPS?

Reply/change: We thank Reviewer 3 for this question. In the original manuscript, we performed solid-state ^{13}C NMR measurements to verify the chemical composition of the PHT aerogels. These results showed that all generations of PHT aerogels were free of side products or impurities, indicating that the reversible nature of the PHT network does not lead to the formation of irreversible structures. We fully agree with Reviewer 3 that under hot-pressing conditions, material degradation due to heat and/or pressure could potentially result in side reactions or by-products. To address this concern, we have conducted additional solid-state ^{13}C MAS NMR experiments on reprocessed thermosets PHT-E1 and PHT-E2. As shown in **Extended data 6a** of the revised manuscript, the chemical shifts of PHT-E1 and PHT-E2 are in line with those of the pristine PHT aerogels, confirming that the chemical structure is retained and no significant side products are formed during reprocessing.

In addition, we also performed FTIR analysis on all PHT aerogels and thermosets. All samples display similar absorbance spectra, indicating no significant changes in functional groups during the recycling process (**Figure R5**). However, we recognize that FTIR alone does not provide conclusive evidence for side products. For instance, hemiaminal intermediates, which may form during PHT synthesis, exhibit a characteristic peak near 3500 cm^{-1} (-OH stretching

region) that is obscured in this spectra. In contrast, solid-state NMR can clearly resolve the presence or absence of such impurities by distinguishing them from the characteristic peaks of the PHT framework.

For this reason, we have emphasized the solid-state NMR results of PHT-E1 and PHT-E2 in the revised manuscript and have chosen to present these data as the primary evidence supporting the chemical integrity of the recycled materials (**Extended data 6a** and page 11, line 20–23).

Figure R5. FTIR spectra of PHT aerogels and thermosets.

5. What are the influence patterns of other types of amine substances on the efficiency of ASA and product performance? Can it replace the above-mentioned aniline raw materials?

Reply: We thank Reviewer 3 for this question. In principle, other primary amines or aniline derivatives could indeed be employed as depolymerization agents in the ASA process. However, in this study, we intentionally selected aniline components that yield fully aromatic PHT materials, which exhibit enhanced thermal stability and mechanical performance compared to partially aromatic or aliphatic analogues.⁴ While the use of alternative amines could certainly be explored in future studies to further broaden the versatility of this system, we believe our choice of amines achieves an effective balance between depolymerization efficiency and the desirable structural and thermal properties demonstrated in this work.

6. The SEM image only shows the local morphology. It is recommended to add TEM or gas pore size distribution map (such as BJH method) to describe the microstructure of the above sample more comprehensively.

Reply/change: We thank Reviewer 3 for this suggestion. We agree with Reviewer 3 that one SEM image cannot fully display the overall morphology of the aerogel network. In response to this point, and in line with Reviewer 2's suggestion, we have revised the manuscript to include two representative SEM images for each sample. As shown in **Figure 2f**, the left panel presents higher-magnification images highlighting the local morphology of the PHT aerogels, while the right panel provides lower-magnification images to show the overall surface structure of the aerogel network.

Furthermore, for samples PHT-B, PHT-C, and PHT-D, we have included pore size distribution data obtained from BET analysis to demonstrate the mesoporous characteristics of the different reprogrammed aerogels (see **Extended data 5b**). We believe that the addition of these SEM images and pore size distribution graphs provides a more comprehensive and convincing

visualization of the PHT aerogel network and clearly illustrates the structural consistency across different samples.

7. NMP is a known hazardous solvent, and its reuse has raised concerns about sustainability. Please discuss potential alternatives or provide reasons for choosing NMP in the context of green chemistry.

Reply: We thank Reviewer 3 for this comment regarding the role of solvents and the sustainable aspects of our synthesis approach. Solvents play an important role in constructing the aerogel network. Previous studies have demonstrated that the nanomorphology of organic aerogels can be significantly influenced by the choice of solvent.⁵ In our prior work, we showed that PHT aerogels prepared using BAPP as the amine source and NMP as the solvent exhibited the best thermal insulation performance.⁶ For this reason, NMP was selected as the primary solvent in the present study. Additionally, we have previously demonstrated that recyclable PHT aerogels can also be prepared using green solvents, further supporting the versatility of our approach.⁷

We fully acknowledge that for overall sustainability, especially within the framework of green chemistry, the recovery and reuse of NMP, which is classified as a hazardous solvent, are crucial. In response to question 3 of Reviewer 3, we have recovered all solvents used during PHT-pristine synthesis with high purity and high recovery yields.

Change: We have revised the manuscript to include a detailed description of the recovery of all solvents used during the PHT-pristine synthesis (see page 5, line 2–4). We also added details of the solvent recovery process and its results in the Supporting Information.

8. Many material property data (for example, surface area, thermal conductivity) have no error lines or statistical significance. Please add the standard deviation and repeat the measurement where applicable.

Reply/Change: We thank Reviewer 3 for this suggestion. In response to Reviewer's request, we added standard deviation data to several material properties of PHT aerogels, including skeletal densities and thermal conductivities (see **Table S2**, **Table S4**, and **Table S5**). To show the statistical significance of our materials, we performed two additional BET measurements of PHT-A3, in which it shows similar specific surface area and mesopore volume values (see **Table S4**).

9. Although the thermal conductivity of aerogels is low, does its performance decline in a humid environment? Its long-term stability under humidity, mechanical cycling or ultraviolet radiation should be evaluated. At least the moisture resistance after multiple cycles should be tested and reported.

Reply: We thank Reviewer 3 for this valuable suggestion. In the manuscript, we have demonstrated that our PHT aerogels exhibit hydrophobic properties, as confirmed by contact angle measurements. We also performed a water uptake test, which showed that even under harsh immersion conditions, the aerogels show low water uptake value, demonstrating their hydrophobic nature. We fully agree with Reviewer 3 that PHT aerogels could potentially experience performance degradation when subjected to long-term exposure to harsh environmental conditions. To address this concern, we conducted additional durability testing by placing PHT-pristine samples in a climate chamber at 70 °C and 70% relative humidity for

two weeks. Following this treatment, we measured both the thermal conductivity and mechanical properties of the aged samples.

As shown in **Figure R6**, the stress–deformation curves of PHT-pristine before and after climate chamber exposure are nearly identical, indicating that the aerogels maintain their mechanical robustness even after extended exposure to elevated temperature and humidity. Furthermore, the thermal conductivity of the PHT-pristine samples over time remained consistent with the initial values, confirming that the thermal insulation performance is also preserved under these conditions (**Table R6**).

Change: We included these experiments along with **Figure R6** and **Table R6** in the revised supporting information (see **Figure S10** and **Table S15**).

Figure R6. Stress-deformation curves of PHT-pristine after humidity treatment.

Table R6. Thermal conductivities of PHT-pristine after humidity treatment

Name	Thermal conductivity, λ [$\text{Wm}^{-1}\text{K}^{-1}$]
PHT-pristine_0 day ^{a)}	0.0189
PHT-pristine_9 days	0.0187
PHT-pristine_14 days	0.0187

^{a)}PHT-pristine_0 day was prepared according to pristine polyhexahydrotriazine aerogel synthesis. The sample was placed in climate chamber at 70 °C and RH 70% for periods of 9 and 14 days, respectively, and the thermal conductivity was recorded.

10. The article points out that the thermal conductivity and mechanical strength can be adjusted by different aromatic amines, but the correlation between the microscopic pore structure and the molecular structure has not been established in detail. Why can the spiro structure of the FDA increase the specific surface area? Can the structural differences of pores at different scales be verified through PALS, BET or SAXS systems?

Reply: We thank Reviewer 3 for this question. The smaller pore structural feature of PHT-B can be attributed to the molecular structure of FDA, which contains a spiro-carbon moiety that

hinders efficient packing during polymerization, resulting in the formation of a microporous structure.⁸ In addition, the incorporation of the rigid, non-coplanar spiro structure disrupts efficient packing during polymerization, generating additional free volume and a more tortuous pore network.²

To investigate this, we have included the pore size distribution profiles of PHT-B and PHT-pristine from the BET measurement (**Figure R3**). As shown in **Figure R3**, PHT-B possesses a higher pore volume in the 10–50 nm range compared to PHT-pristine, supporting the enhanced mesoporosity introduced by the spiro structure.

Change: We added detailed explanation on the microporous characteristic of PHT-B (see page 8, line 28–35 and page 9, line 1–2). We also included the pore size distribution profile of reprogrammed PHT aerogels in the revised manuscript as **Extended data 5b**.

11. At present, new materials are synthesized only based on different amines, but do different crosslinking degree networks have an inhibitory effect on the reactivity of ASA? A denser network may lead to difficulties in disaggregation.

Reply: We thank Reviewer 3 for this question. The ASA process relies on the decrosslinking of the PHT network through aminolysis. As the crosslink density of the aerogel network increases, it could become more challenging to achieve complete dissolution of the resulting oligomers. To investigate this, we selected PHT-D as our starting material because its embedded hexafunctional amines increase the overall crosslink density (**Figure R7**).

To test the depolymerization, we added a 14 wt% BAPP solution in NMP, and the mixture was ultrasonicated at 65 °C for 4 h. After sonication, a clear dissolution of the aerogel was observed. The resulting depolymerized mixture was first analyzed by NMR spectroscopy. In the ¹H NMR spectrum, characteristic signals for aminal structures were detected at 4.4 and 6.3 ppm (**Figure R8a**). Additionally, the ¹H COSY NMR spectrum confirms that these two signals are coupled, providing further evidence for the presence of the embedded aminal structures (**Figure R8b**).

Moreover, the DEPT ¹³C NMR spectrum shows a methylene carbon signal at 54 ppm, corresponding to the aminal groups, similar to the original system described in the manuscript. To confirm the presence and size of the formed oligomers (**Figure R8c**), ¹H DOSY NMR experiments were conducted. The spectra revealed that the oligomeric structures exhibiting the characteristic aminal signals are indeed larger in size than the BAPP amine (**Figure R8d**). Overall, these findings are consistent with our previous depolymerization results and indicate that the aminolysis reaction remains effective even in systems with higher crosslink density.

However, we decided to not include this finding in the revised manuscript, as our primary aim is to showcase the ASA process through the reactivity of PHT. Therefore, for consistency and clearer comparison, we chose to focus on a single type of PHT aerogel network with a uniform crosslink density, namely, PHT-pristine.

Figure R7. a) Photographs displaying the protocol of aminolysis of PHT-D. PHT-D powders were treated with a 14 wt% BAPP solution in NMP, and the mixture was ultrasonicated for 4 h until full dissolution. The clear red aerogel dissolution can be obtained.

Figure R8. NMR spectroscopy experiment demonstrating the partially depolymerization of PHT-D. a) ^1H NMR spectrum (400 MHz, 25 °C, $\text{DMSO-}d_6$), b) ^1H COSY NMR spectrum (400 MHz, 25 °C, $\text{DMSO-}d_6$), c) DEPT ^{13}C NMR spectrum (100 MHz, 25 °C, $\text{DMSO-}d_6$), and d) ^1H DOSY NMR spectrum after dissolution of PHT-D. The signals of different compounds were marked in different colors (red: BAPP and blue: oligomers).

References

1. Koebel, M., Rigacci, A. & Achard, P. Aerogel-based thermal superinsulation: An overview. *Journal of Sol-Gel Science and Technology* vol. 63 315–339 Preprint at <https://doi.org/10.1007/s10971-012-2792-9> (2012).

2. Rashidi, V., Coyle, E. J., Sebeck, K., Kieffer, J. & Pipe, K. P. Thermal Conductance in Cross-linked Polymers: Effects of Non-Bonding Interactions. *Journal of Physical Chemistry B* **121**, 4600–4609 (2017).
3. Vert, M. *et al.* Terminology for biorelated polymers and applications (IUPAC recommendations 2012). *Pure and Applied Chemistry* **84**, 377–410 (2012).
4. Kaminker, R. *et al.* Solvent-Free Synthesis of High-Performance Polyhexahydrotriazine (PHT) Thermosets. *Chemistry of Materials* **30**, 8352–8358 (2018).
5. Taghvaei, T. *et al.* K-Index: A Descriptor, Predictor, and Correlator of Complex Nanomorphology to Other Material Properties. *ACS Nano* **13**, 3677–3690 (2019).
6. Wang, C.-L., Chen, Y.-R., Eisenreich, F. & Tomović, Ž. One-Step Synthesis of Closed-Loop Recyclable and Thermally Superinsulating Polyhexahydrotriazine Aerogels. *Advanced Materials* **2412502**, 1–9 (2024).
7. Wang, C. L., Glišić, I., Chen, Y. R. & Tomović, Ž. Closed-Loop Recyclable Polyhexahydrotriazine Aerogels Utilizing N,N-Dimethyl Lactamide as a Green Solvent. *ChemSusChem* (2025) doi:10.1002/cssc.202500125.
8. Weber, J., Su, Q., Antonietti, M. & Thomas, A. Exploring polymers of intrinsic microporosity - Microporous, soluble polyamide and polyimide. *Macromol Rapid Commun* **28**, 1871–1876 (2007).

Manuscript number: NCOMMS-25-23496-T

Title: " Advancing Aerogel Recyclability through Polyhexahydrotriazine Reactivity"

Comments to the Editor:

We would like to express our sincere gratitude to the editorial team and reviewers for their time and effort in evaluating our manuscript. We greatly appreciate the thoughtful feedback and insightful suggestions provided throughout the review process.

In response to the editorial guidance and the final reviewer comments, we have carefully revised the manuscript to align with the standards and expectations of *Nature Communications*. Specifically, we have addressed the editorial requests outlined in the Author Checklist and incorporated the final recommendation regarding the use of the term “thermoset.” As suggested, we have revised the language to describe our materials as “thermoset-like films” or “cross-linked polymeric film”.

We have also updated the manuscript accordingly, with all changes highlighted in yellow for ease of review. We believe these final revisions fully address the remaining concerns and further strengthen the clarity and impact of our work. We hope the revised version meets the journal’s requirements and will be of broad interest to the *Nature Communications* readership.

Point-by-point reply to the reviewers

Reviewer #2 (Remarks to the Author):

I thank the authors for their extremely diligent and thorough job in addressing my comments and revise the manuscript. All my concerns, except one, have been fully addressed and for all practical purposes the manuscript is acceptable for publication. My only remaining concern is related to the use of the term “thermoset.”

I had commented as follows (bold facing was added now for quick referencing):

The most serious, in my opinion, issue lies in the characterization of the hot-pressed films as “thermosets.” Based on both experimental data and standard polymer definitions, this classification is not justified:

True thermosets are irreversibly cured materials that cannot be reshaped or reprocessed. This typically results from heat-induced crosslinking that forms a permanent, tightly bonded network.

By contrast, the PHT-derived films are chemically identical to the pre-hot-pressed samples, and therefore can be dissolved in the presence of primary amines and re-gelled into aerogels that are chemically and mechanically indistinguishable from the pristine material.

The reported insolubility of the hot-pressed films is not caused by hot-press-induced crosslinking. Rather, it is an inherent property of the pre-existing PHT network, which, by its nature, is already crosslinked. There is no evidence of new, irreversible curing during hot pressing.

The Authors responded as follows (bold facing and underlining were added now for quick referencing):

*More importantly, the reprocessed thermoset presented in this work possesses inherent recyclability, enabling its transformation into next-generation high-performance aerogels, which is the primary focus of this research. **We agree with the reviewer's definition of a true thermoset as an irreversibly cured material that cannot be reshaped or reprocessed. The PHT network, by contrast, is inherently reversible under appropriate conditions (e.g., in the presence of primary amines) that can transformed back to organic aerogels.** The PHT thermosets still possess strong mechanical/thermal properties as conventional thermosets. In this light, we decided to keep the term of thermosets in the manuscript, which we believe appropriately conveys our material.*

“Thermoset” is a formal term that, according to IUPAC, is defined as: “*A polymer that becomes irreversibly hard on heating. A thermoset does not soften or melt on reheating, but instead may decompose.*” Although I understand the author’s point of view (underlined part of their response), I am not sure whether *Springer Nature* allows colloquial use of formal terminology. I invite the editorial team to arbitrate this point.

Finally, I would suggest replacing the word “Undiscovered” in the revised title with something more tangible, such as “Reversible.” As for example: **Advancing Aerogel Recyclability via Reversible Polyhexahydrotriazine Chemistry.**

Reply: We thank Reviewer 2 for their thoughtful and constructive comments, and for acknowledging our revisions. We understand the Reviewer’s continued concern regarding the use of the term “thermoset” to describe our hot-pressed PHT-derived films. As correctly noted by the Reviewer, conventional thermosets are defined as irreversibly cured polymers that do not melt or dissolve upon reheating. We fully agree with the Reviewer that the recyclability of the PHT network under specific chemical conditions (i.e., exposure to primary amines) distinguishes it from conventional thermosets. To better reflect this distinction, we have revised the manuscript to more explicitly describe the material as a “thermoset-like films” or “cross-linked polymer, which properly described our recyclable PHT materials. We hope this phrasing addresses the Reviewer’s concern.

In response to the Reviewer’s suggestion regarding the manuscript title, we followed the suggestions of Editor and have revised the title as “Advancing Aerogel Recyclability through Polyhexahydrotriazine Reactivity. All changes related to terminology and title have been highlighted in yellow in the revised manuscript. We believe the revised version fully addresses the Reviewer’s comments and concerns.

Overall

The authors expand the concept of Dynamic Covalent Chemistry (DCC) by introducing a new member to the typical list of reversible reactions, which includes imine formation/hydrolysis, disulfide exchange, boronate ester exchange, and Diels–Alder reactions. The new addition is the hexahydrotriazine (HT) 6-membered ring, which is formed via the condensation of primary amines and formaldehyde. The authors discovered that this structure can be broken down into a mixture of soluble oligomers by adding an excess of a primary amine and subsequently reformed by reintroducing the missing stoichiometric amount of formaldehyde. In turn, polyhexahydrotriazine (PHT)-based wet gels and aerogels are synthesized using diamines.

This newly discovered chemistry enables waste-free chemical recycling of organic aerogels, which are valuable as thermal insulators. Notably, by varying the amine feedstock during depolymerization, the regenerated aerogels can possess different, tailored properties compared to the original material.

Furthermore, it is reported that the application of heat and pressure to induce HT bond exchange converts aerogels into high-performance thermosets, which can then be reverted into aerogels. However, as I will elaborate below, I find the use of the term *thermoset* problematic in this context.

The paper begins with an **Introduction** that clearly outlines the motivation for the study and situates the new materials and processes within the broader context of chemical transformations and aerogel science. The **Results and Discussion** section is divided into four parts:

- (a) An examination of the dynamic behavior of HT structures;
- (b) A description of the synthesis, properties, and closed-loop recycling of PHT aerogels;
- (c) Reprogramming of PHT aerogels; and,
- (d) Reprocessing of PHT aerogels.

The paper concludes with a **Conclusions** section, which is mistakenly labeled “Discussion.”

Comments on Specific Sections – Strengths and Weaknesses

Introduction

The Introduction clearly, accurately, and concisely traces the intellectual evolution of polymer recycling—from the conventional approach of hydrolyzing polymers into monomers, which is often wasteful, to the more efficient strategy of breaking down $(-A-B-)_n$ polymers (in aerogel form) into soluble oligomers (e.g., A-B-A or B-A-B) through reaction with one of the monomers (A or B). The polymer network can then be regenerated by adding the stoichiometric amount of the missing monomer (B or A). This reversible process, referred to as the *aerogel-to-sol-to-aerogel* (ASA) cycle, is powerful because it avoids complete depolymerization into monomers and their subsequent isolation. The present work extends this approach to a new class of aerogels based on PHT.

Comments for Improvement

The opening sentence - “*Organic aerogels, identified by IUPAC as one of the top ten emerging technologies...*” - is inaccurate. The relevant IUPAC report refers to *aerogels* in general, not exclusively *organic aerogels*. Therefore, it would be more appropriate to revise the opening to:

“Aerogels, identified by IUPAC as one of the top ten emerging technologies...”

The following sentence, “*These materials exhibit ultralow thermal conductivity... 0.020 W m⁻¹ K⁻¹...*”, contains an incorrect value. Inorganic aerogels (e.g., silica) have reported thermal conductivities as low as 0.012 W m⁻¹ K⁻¹, and even organic aerogels have achieved values of 0.018 W m⁻¹ K⁻¹ or lower. This figure should be updated for accuracy. Furthermore, a clearer sentence structure would be:

“Organic aerogels in particular have shown promise due to their low density and mechanical tunability...”

Rewording Recommendation (End of Page 2):

Approximately seven lines from the bottom of page 2, the sentence could be rephrased for clarity and impact as:

“...the material can be repurposed in every cycle on demand to enhance the thermal conductivity, mechanical strength, or thermal/flame resistance of the next-generation aerogel.”

Clarification Needed (Four Lines from Bottom of Page 2):

The statement “*These thermosets demonstrate improved thermal and mechanical performance...*” lacks a clear point of comparison. Improved relative to what? The original aerogels? Previously reported thermosets? Moreover, as discussed later in conjunction to Part c, the concept of “thermosets” produced via “compression molding” from dynamic covalent polymers should be reconsidered or postponed for a future publication.

Figure 1 – Visual and Informational Improvements:

- **Font Size:** The font in Figure 1 is too small and difficult to read. All labels should be increased in size for legibility.
- **Contrast Issues:** White font over colored backgrounds suffers from low contrast. Please consider using darker font or adjusting background shades.
- **Clarity of Cartoons:** Several cartoon illustrations in Figure 1b are too small and low in contrast to be interpreted effectively.
- **Missing Label:** Please indicate the solvent used (NMP) under the first arrow in Figure 1b.

Results and Discussion

Part (a): Dynamicity of the HT Structure

Using a model reaction involving a mono-primary amine and paraformaldehyde, the authors provide experimental evidence for the formation of the HT ring. This is followed by depolymerization through excess primary amine and quantitative regeneration of the HT structure upon the reintroduction of paraformaldehyde in the missing stoichiometric amounts.

Additionally, the authors present evidence of a **self-exchange reaction** - although they do not use this term - among the substituents on the nitrogen atoms of the HT ring under heat and pressure. The term *metathesis* is employed instead, likely due to the experimental design comparing methoxyphenyl and ethoxyphenyl substituents and observing scrambling in the resulting HT structures. While technically correct, the term “self-exchange” may also more clearly communicate the underlying dynamic mechanism.

This last discovery is highly significant as it demonstrates that HT linkages maintain their reversibility under thermal and mechanical stress - mirroring solution-phase dynamicity. However, the broader implications of this behavior are not fully explored in the paper. Given this, I remain skeptical on whether **Part (d): Reprocessing of PHT Aerogels** is adequately developed. As it will be discussed below, it may be better to reserve this part for a future, more in-depth study or significantly revised to include expanded analyses.

Technical and Conceptual Notes on Hot-Pressing in Part (a)

1. Units Correction:

The paper states:

“A model experiment was conducted ... The process involved applying 30 kN of pressure at 180 °C for 0.5 h, followed by 100 kN of pressure at 180 °C...”

This is a misuse of units. Kilonewtons (kN) are units of force, not pressure. The pressure applied should be expressed in units of **force per area** (e.g., MPa or bar).

2. Interpretation of Structural Integrity and MALDI-TOF Results:

The authors conclude that the HT structure shows “no signs of degradation” under heat and pressure, as evidenced by consistent ¹H NMR spectra. However, the MALDI-TOF analysis shows that substituent scrambling occurred - **an indication of dynamic bond exchange, not static structural integrity**.

Therefore, the interpretation should be revised. Rather than affirming structural integrity, the data more accurately suggest **quantitative dynamic behavior** with no apparent degradation or irreversible side reactions - an important and promising result. This should be framed as:

“These findings indicate robust dynamicity under thermal and mechanical conditions, comparable to that observed in solution, without evidence of irreversible degradation.”

Part (b): Synthesis and Closed-Loop Recycling of PHT Aerogels

This section represents the logical continuation of the findings in Part (a), translating the demonstrated dynamicity of the HT system into a practical material - namely, a PHT aerogel. The authors achieve this by reacting formaldehyde with a primary diamine, forming a wet gel that undergoes gelation, controlled degradation, and successful regelation across two subsequent cycles. The polymerization and depolymerization processes are monitored using a comprehensive suite of analytical techniques, and the experimental design is clear and methodically executed.

The presentation is concise yet compelling. Impressively, the results show that the key material properties—bulk density, porosity, specific surface area, thermal conductivity, and water uptake—are nearly identical across the pristine aerogel and its first and second regenerated forms. While the manuscript does not use the term *reconstituted aerogels*, this terminology is highly accurate and effectively captures the process of iterative recycling while retaining material performance. The authors are encouraged to adopt this term in future revisions.

Technical Comments and Clarifications:

- **Terminology Correction (Page 5, Top):**

The manuscript states:

“The PHT aerogel (PHT-pristine) exhibits low bulk density (0.15 g cm^{-3}) and high porosity (88%), key features of lightweighted porous materials.”

The term *lightweighted* is incorrect. It should be replaced with *lightweight*, which is the proper adjective.

- **Surface Area and Pore Volume Misinterpretation:**

The authors report:

“PHT-pristine also shows specific surface area of $129 \text{ m}^2 \text{ g}^{-1}$ and a pore volume of $0.25 \text{ cm}^3 \text{ g}^{-1}$...”

However, based on the reported bulk density ($\rho_b = 0.15 \text{ g cm}^{-3}$) and porosity ($\Pi = 88\%$), the **skeletal density** (ρ_s) is calculated as: $\rho_s = 1.25 \text{ g cm}^{-3}$

Using this, the **total pore volume** (V_{Total}) can be calculated as:

$$V_{\text{Total}} = (1/\rho_b) - (1/\rho_s) = 5.87 \text{ cm}^3 \text{ g}^{-1}$$

Therefore, the reported pore volume of $0.25 \text{ cm}^3 \text{ g}^{-1}$ **reflects only the mesoporous fraction accessible via nitrogen sorption analysis**. The actual pore architecture is predominantly **macroporous**, as also indicated by the average pore diameter (Φ), calculated using:

$$\Phi = 4 \times V_{\text{Total}} / (\text{specific_suf._area}) = 182 \text{ nm}$$

Consequently, the claim of a *mesoporous architecture* is misleading and should be rephrased to reflect the macroporous character of the material.

- **Thermal Conductivity Analysis:**

From the porosity ($\Pi = 0.88$) and the average pore diameter ($\Phi = 182 \text{ nm}$), one can estimate the gaseous thermal conductivity as approximately $9.1 \text{ mW m}^{-1} \text{ K}^{-1}$, down from $\sim 26 \text{ mW m}^{-1} \text{ K}^{-1}$ in open still air, confirming that the Knudsen effect significantly reduces gas-phase conduction. However, (a) as data show the Knudsen effect is not a purely mesoporous effect; and (b) the remaining $\sim 9.9 \text{ mW m}^{-1} \text{ K}^{-1}$ in the total thermal conductivity must be attributed to solid-phase conduction through the polymer framework. This nuance should be emphasized to avoid oversimplifying the origin of the thermal insulation property of aerogels as a purely mesoporous phenomenon.

Figure Recommendations:

- **SEM Imaging:**

All SEMs presented in Figures 2 and 3 would benefit from dual magnification, with one half at low magnification for overall morphology and the other at high magnification to show fine features (see for example N. Leventis' publications for reference).

- **Figure Formatting:**

Please apply the same feedback regarding font size, contrast, and label clarity from Figure 1 to Figures 2 through 5. Specifically, all text should be legible without magnification, and white text on colored backgrounds should be adjusted for improved readability.

An Important Conceptual Highlight from Part b:

Before closing by discussion of Part b, I feel compelled to comment that a particularly remarkable point from this section is that **well-defined monomers are only used in the initial synthesis**. In subsequent cycles, following *disintegration* of the first-generation aerogel, the reactive feedstock becomes a **solution of oligomers**. Despite the lack of defined monomer units, or precise knowledge about the chemical identity of the oligomers, this mixture behaves functionally as if it were composed of pure monomers, yielding second- and third-generation aerogels that are **practically indistinguishable from the original** in terms of physical properties.

This observation motivates a more precise use of terminology: whenever possible, *disintegration* is preferable to *depolymerization*, as the latter may imply full reversion to monomeric species, which is not the case here. This distinction is important for accurately describing the chemical process at work.

Part (c): Reprogramming of PHT Aerogels

This section introduces a powerful and forward-thinking capability of the PHT dynamic network: **property reconfiguration through selective amine substitution** during the disintegration and reconstitution process. By using different primary diamines in place of the original ones during depolymerization, the authors demonstrate that each new generation of the PHT network can be tailored for distinct applications - many of which were not envisioned in the original aerogel synthesis.

This approach constitutes a form of chemical reprogramming, enabling the rational design of materials with specific functionalities, all while preserving the core aerogel structure and recyclability.

Case Studies in Amine Substitution:

1. **Rigid Diamine with Spiro-Carbon Moiety:** The incorporation of a rigid diamine containing a spiro-carbon motif results in a stiffer molecular backbone. This rigidity translates to microporous aerogels with enhanced specific surface area and lower thermal conductivity, likely due to increased phonon scattering and better suppression of heat transfer pathways. This transformation exemplifies how structural features at the molecular level can cascade into macroscopic performance changes.
2. **Diamine with Dual Amide Groups:** In this case, the selected diamine includes and therefore introduces two amide groups capable of forming interchain hydrogen bonds (of Kevlar type). The resulting second-generation aerogels display enhanced mechanical strength relative to the original PHT-pristine material. This outcome highlights the use of non-covalent secondary interactions to bolster network cohesion without altering the primary covalent backbone - a good strategy for strengthening dynamic networks.
3. **Hexafunctional Amine with Phosphazine Core:** The use of a multifunctional amine incorporating a phosphazine ring imparts significant thermal stability and flame resistance to the reconstituted aerogels. This formulation illustrates the versatility of the system in integrating specialty performance features through feedstock selection alone, without modifying the base chemistry or synthesis route.

Evaluation and Suggestions:

This section is both **innovative and well-supported**. It demonstrates how a single dynamic polymer network architecture can be **reprogrammed on demand** to meet diverse functional requirements—a hallmark of next-generation sustainable materials.

That said, several opportunities for enhancement exist:

- **Comparative Data Presentation:** A similar analysis of the total thermal conductivity to that presented in Part b in terms of the gaseous and solid contributions would be helpful in assessing the effect of the molecular rigidity in the case of the rigid diamine with a spiro-carbon moiety. It will be very instructive to see how much of the lowering in the thermal conductivity ($15.9 \text{ mW m}^{-1} \text{ K}^{-1}$ vs. $19 \text{ mW m}^{-1} \text{ K}^{-1}$) can be attributed to the rigidization of the skeletal framework brought about by the spiro structure.
- **Mechanistic Insights:** A more detailed discussion of how each structural feature (e.g., spiro-centers, amide linkages, phosphazine rings) affects the network architecture and physical properties would strengthen the section's depth.
- **Broader Context:** The reprogramming concept would benefit from a brief connection to related work in the dynamic covalent chemistry literature - particularly in the context of adaptive materials.

Overall, this section solidly reinforces the adaptability of the PHT platform and extends its potential far beyond traditional monomer-based material design. The ability to repurpose recycled materials into enhanced or application-specific forms is a major sustainability advance and should be emphasized as such in the manuscript.

Part (d): Reprocessing of PHT Aerogels

In this section, the authors explore **thermal and mechanical reprocessing** of HT-based aerogels. Specifically, aerogel pieces with the same composition as described in Part (b) were **hot-pressed** - as initially outlined in Part (a) - to produce **substantially nonporous, transparent films**. These materials are referred to as *thermosets* and were characterized in terms of their mechanical and

thermomechanical properties, including solvent resistance. The films were found to be insoluble in several common organic solvents.

Subsequently, these hot-pressed films were redissolved in NMP (N-methyl-2-pyrrolidone) containing the same primary diamine monomer originally used in their synthesis (described in Part b). Upon the addition of the missing stoichiometric amount of formaldehyde, the PHT network was regenerated, resulting in aerogels that were recycled, hot-pressed, and reprocessed again. Across these cycles, the authors report consistent values for key metrics - bulk density, porosity, specific surface area, water contact angle, microstructure, and thermal conductivity - as well as comparable mechanical properties. These similarities are presented as evidence that:

“...the polymer structure of PHT-Fs remains mechanically robust even after undergoing heat and pressure treatment.”

Critical Evaluation: Conceptual Concerns with Part (d):

While this section is mostly technically sound, I have reservations about its contribution to the overall narrative of the paper. Up to this point, **Parts (a) through (c) build momentum** through increasingly impactful demonstrations of the dynamic capabilities of the HT/PHT platform. **In contrast, Part (d) feels like a conceptual regression - a shift into lower gear, if you wish.**

Since hot-pressing does not degrade the HT system (as shown by the control experiments in Part a), the solution-phase chemistry of the hot-pressed aerogels is, as expected, identical to that of the original PHT materials. Thus, it could be argued that there is little new insight gained from this section. If the intent was to demonstrate that PHT aerogels can be reconfigured as dense plastics, that is valid, but not particularly unique to the PHT aerogels or relevant to this manuscript.

The most serious, in my opinion, issue lies in the characterization of the hot-pressed films as “thermosets.” Based on both experimental data and standard polymer definitions, this classification is not justified:

- True thermosets are irreversibly cured materials that cannot be reshaped or reprocessed. This typically results from heat-induced crosslinking that forms a permanent, tightly bonded network.
- By contrast, the PHT-derived films are chemically identical to the pre-hot-pressed samples, and therefore can be dissolved in the presence of primary amines and re-gelled into aerogels that are chemically and mechanically indistinguishable from the pristine material.
- The reported insolubility of the hot-pressed films is not caused by hot-press-induced crosslinking. Rather, it is an inherent property of the pre-existing PHT network, which, by its nature, is already crosslinked. There is no evidence of new, irreversible curing during hot pressing.

In this context, the repeated assertion that the polymer structure remains unchanged after heat and pressure - while highlighting the robustness of the network - also undermines the novelty of the section. The chemical behavior is unchanged, and therefore the added value of this part is limited.

Alternative and More Impactful Experimental Directions:

Rather than revisiting dynamics already demonstrated in solution (Part b), this section could have explored a solid-state reprogramming strategy, which would better align with the transformative scope of the paper - and with the model hot-pressing experiments from Part (a):

- **Scenario 1:** Hot-press a pristine aerogel in the presence of a **different diamine** to trigger **amine exchange** under heat and pressure. This could enable **solid-state reconfiguration** of the polymer network and aerogel properties.
- **Scenario 2:** Extend this by hot-pressing the aerogel with both a **new diamine and paraformaldehyde**, attempting a **full solid-state network reconstruction**. Though ambitious (read: a long shot), this could eliminate solvent use entirely and further enhance the sustainability profile of the system.

Either approach would provide an opportunity to expand the principles established earlier into a **new application phase**, adding **originality and technical depth** to the manuscript.

Conclusions

The Conclusions section effectively summarize the key innovations of the study. However, the statement:

“Beyond aerogels, the dynamic characteristics of PHT allowed us to fabricate high-performance thermosets from existing aerogels and vice versa”

is misaligned with the evidence provided. Given that the so-called “thermosets” remain fully re-processable, chemically identical to the pristine samples, and lack irreversible curing, this statement overreaches and risks diminishing the technical rigor of the conclusions drawn from Parts (a)–(c).

Final Recommendation

Despite the concerns outlined regarding Part (d), this manuscript presents a highly original, technically sound, and impactful contribution. The demonstration of dynamic covalent behavior, closed-loop aerogel recycling, and chemical reprogramming of material properties represents significant progress in the field of sustainable porous materials.

I support publication pending revisions that:

- Correct or remove the "thermoset" terminology from the Title of the article and claims in Part (d);
- Consider replacing Part (d) with a more forward-looking experiment; and
- Adjust the Conclusions to accurately reflect the scope and novelty of the demonstrated results.

The manuscript presents an innovative and potentially impactful study on the dynamic covalent behavior of hexahydrotriazine (HT) and its application in fully recyclable aerogels and thermosets. The discovery that HT units can undergo transamination and metathesis reactions under mild conditions opens up new avenues for designing circular polymeric materials. The proposed aerogel-to-sol-to-aerogel (ASA) strategy is compelling and aligns well with the current demand for sustainable materials. However, while the manuscript demonstrates a high level of technical rigor and novelty, several key aspects require clarification, additional data, and better articulation before it can be considered for publication. I recommend major revision.

1. The manuscript claims unprecedented dynamicity of HT structures, but lacks detailed mechanistic discussion. The authors should provide deeper insights—ideally supported by theoretical calculations (e.g., DFT)—into the activation energy, reaction pathway, and reversibility under realistic conditions.

2. Only two ASA recycling cycles are presented. For a truly circular material platform, a more extensive demonstration (e.g., ≥ 10 cycles) is necessary to validate long-term recyclability and performance retention.

3. Although the authors claim "waste-free" and "100% of the initial chemical resources remained," no quantitative mass balance or recovery yield data are provided. Please include such data to support this strong claim

4. Although the complex reaction between HTs is considered to be reversible, are there irreversible structures such as secondary condensation and cross-linking densification (e.g., C-C bonding or excessive cross-linking) under hot-pressing conditions, and can these structures be recovered by ASA? Can the authors further confirm this by techniques such as FTIR and XPS?

5. What are the influence patterns of other types of amine substances on the efficiency of ASA and product performance? Can it replace the above-mentioned aniline raw materials?

6. The SEM image only shows the local morphology. It is recommended to add TEM or gas pore size distribution map (such as BJH method) to describe the microstructure of the above sample more comprehensively.

7. NMP is a known hazardous solvent, and its reuse has raised concerns about sustainability. Please discuss potential alternatives or provide reasons for choosing NMP in the context of green chemistry.

8. Many material property data (for example, surface area, thermal conductivity) have no error lines or statistical significance. Please add the standard deviation and

repeat the measurement where applicable.

9. Although the thermal conductivity of aerogel is low, does its performance decline in a humid environment? Its long-term stability under humidity, mechanical cycling or ultraviolet radiation should be evaluated. At least the moisture resistance after multiple cycles should be tested and reported.

10. The article points out that the thermal conductivity and mechanical strength can be adjusted by different aromatic amines, but the correlation between the microscopic pore structure and the molecular structure has not been established in detail. Why can the spiro structure of the FDA increase the specific surface area? Can the structural differences of pores at different scales be verified through PALS, BET or SAXS systems?

11. At present, new materials are synthesized only based on different amines, but do different crosslinking degree networks have an inhibitory effect on the reactivity of ASA? A denser network may lead to difficulties in disaggregation.

I thank the authors for their extremely diligent and thorough job in addressing my comments and revise the manuscript. All my concerns, except one, have been fully addressed and for all practical purposes the manuscript is acceptable for publication. My only remaining concern is related to the use of the term “thermoset.”

I had commented as follows (bold facing was added now for quick referencing):

The most serious, in my opinion, issue lies in the characterization of the hot-pressed films as “thermosets.” Based on both experimental data and standard polymer definitions, this classification is not justified:

- **True thermosets are irreversibly cured materials that cannot be reshaped or reprocessed.** This typically results from heat-induced crosslinking that forms a permanent, tightly bonded network.
- **By contrast, the PHT-derived films are chemically identical to the pre-hot-pressed samples, and therefore can be dissolved in the presence of primary amines and re-gelled into aerogels that are chemically and mechanically indistinguishable from the pristine material.**
- The reported insolubility of the hot-pressed films is not caused by hot-press-induced crosslinking. Rather, it is an inherent property of the pre-existing PHT network, which, by its nature, is already crosslinked. There is no evidence of new, irreversible curing during hot pressing.

The Authors responded as follows (bold facing and underlining were added now for quick referencing):

More importantly, the reprocessed thermoset presented in this work possesses inherent recyclability, enabling its transformation into next-generation high-performance aerogels, which is the primary focus of this research. **We agree with the reviewer’s definition of a true thermoset as an irreversibly cured material that cannot be reshaped or reprocessed. The PHT network, by contrast, is inherently reversible under appropriate conditions (e.g., in the presence of primary amines) that can transformed back to organic aerogels.** The PHT thermosets still possess strong mechanical/thermal properties as conventional thermosets. In this light, we decided to keep the term of thermosets in the manuscript, which we believe appropriately conveys our material.

“Thermoset” is a formal term that, according to IUPAC, is defined as: “A *polymer that becomes irreversibly hard on heating. A thermoset does not soften or melt on reheating, but instead may decompose.*” Although I understand the author’s point of view (underlined part of their response), I am not sure whether *Springer Nature* allows colloquial use of formal terminology. I invite the editorial team to arbitrate this point.

Finally, I would suggest replacing the word “Undiscovered” in the revised title with something more tangible, such as “Reversible.” As for example: **Advancing Aerogel Recyclability via Reversible Polyhexahydrotriazine Chemistry.**